# LOVA³: Learning to Visual Question Answering, Asking and Assessment

**Henry Hengyuan Zhao**[1]**, Pan Zhou**[2†]**, Difei Gao**[1]**, Zechen Bai**[1]**, Mike Zheng Shou**[1†]
[1]Show Lab, National University of Singapore,
[2]Singapore Management University

## Abstract

Question answering, asking, and assessment are three innate human traits crucial for understanding the world and acquiring knowledge. By enhancing these capabilities, humans can more effectively utilize data, leading to better comprehension and learning outcomes. Current Multimodal Large Language Models (MLLMs) primarily focus on question answering, often neglecting the full potential of questioning and assessment skills. Inspired by the human learning mechanism, we introduce **LOVA³**, an innovative framework named "Learning tO Visual question Answering, Asking and Assessment," designed to equip MLLMs with these additional capabilities. Our approach involves the creation of two supplementary training tasks **GenQA** and **EvalQA**, aiming at fostering the skills of asking and assessing questions in the context of images. To develop the questioning ability, we compile a comprehensive set of multimodal foundational tasks. For assessment, we introduce a new benchmark called **EvalQABench**, comprising 64,000 training samples (split evenly between positive and negative samples) and 5,000 validation and testing samples. We posit that enhancing MLLMs with the capabilities to answer, ask, and assess questions will enhance their multimodal comprehension, ultimately improving overall performance. To validate this hypothesis, we train MLLMs using the **LOVA³** framework and evaluate them on a range of multimodal datasets and benchmarks. Our results demonstrate consistent performance gains, underscoring the critical role of these additional tasks in fostering comprehensive intelligence in MLLMs. The code is available at https://github.com/showlab/LOVA3.

## 1 Introduction

To acquire knowledge, we humans often answer lots of questions and then improve ourselves by comparing our answers with the ground-truth answers. As a result, this learning mechanism empowers humans with the answering ability, which allows humans to handle well many real tasks, such as visual question answering [26, 54, 30, 27, 43, 42]. However, as described in the following slogan,

> "The art of proposing a question must be held in higher value than solving it." - Georg Cantor [6]

asking a question is very valuable and even more important than answering a question. Indeed, humans also acquire knowledge from learning to ask questions since it encourages individuals to engage more deeply with information, thereby enhancing problem-solving skills [69, 70, 21, 57]. In addition to asking questions, humans also improve themselves through self-evaluation: humans try to identify the correctness of the answer and thus are involved in a deep understanding of our diverse world [86, 67].

---

[†]Corresponding author.

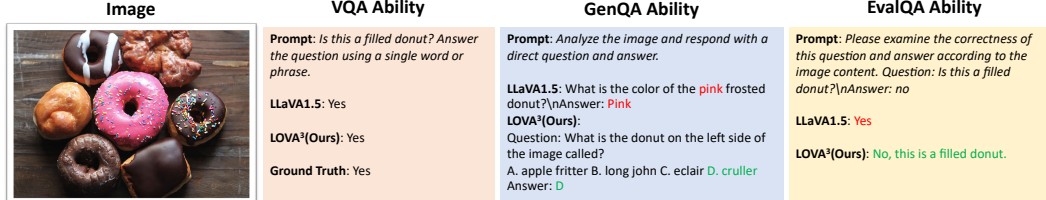

| Image | VQA Ability | GenQA Ability | EvalQA Ability |
|---|---|---|---|

**VQA Ability**

**Prompt**: *Is this a filled donut? Answer the question using a single word or phrase.*

**LLaVA1.5**: Yes

**LOVA³(Ours)**: Yes

**Ground Truth**: Yes

**GenQA Ability**

**Prompt**: *Analyze the image and respond with a direct question and answer.*

**LLaVA1.5**: What is the color of the pink frosted donut?\nAnswer: Pink
**LOVA³(Ours)**:
Question: What is the donut on the left side of the image called?
A. apple fritter B. long john C. eclair D. cruller
Answer: D

**EvalQA Ability**

**Prompt**: *Please examine the correctness of this question and answer according to the image content. Question: Is this a filled donut?\nAnswer: no*

**LLaVA1.5**: Yes

**LOVA³(Ours)**: No, this is a filled donut.

Figure 1: Comparison of three abilities reveals that LLaVA1.5 excels in providing answers but struggles in asking accurate questions and assessing question-answer pairs.

Although innate to humans, apart from answering ability, two other learning mechanisms of asking and assessment remain underexplored for contemporary multimodal large language models (MLLMs). Current MLLMs [42, 84, 3, 95] are excelled in addressing diverse domains of multimodal questions such as mathematics [4], science [42], and commonsense knowledge [17]. However, they predominantly revolve around visual question answering (VQA). As a result, as shown in Fig. 1, current MLLMs, e.g., the representative LLaVA-1.5 [42], suffer from inferior performance on asking questions and self-assess question-answer pairs (QA), which underscores their efficacy as problem-solvers and prohibits holistic multimodal understanding.

To advance the comprehensive intelligence of MLLMs, we introduce two essential tasks: **GenQA** and **EvalQA**, aiming at bolstering the intelligence and robustness of MLLMs. GenQA focuses on enabling the model to generate diverse question-answer (QA) pairs from the single input image, thus equipping the MLLM with the capability to ask questions. We believe that if an MLLM can successfully generate QA pairs for challenging tasks, it indicates a higher level of problem-solving ability [40, 73]. Specifically, we define the GenQA task to include not only generic VQA (e.g., VQAv2 [26] and GQA [30]) but also Multi-Choice VQA (MC VQA), and Multi-Turn VQA (MT) to increase the variety of data formats. Additionally, we incorporate two challenging multimodal grounding tasks into the training process: Referring Expression Comprehension (REC) and Referring Expression Generation (REG). Learning to generate the data of these grounding tasks forces the MLLM to extract fine-grained visual cues from images, such as explicit object localization and compositional relationships. This, in turn, enhances the multimodal reasoning ability of MLLMs. During training, we gather the relevant datasets for these tasks and transform them into a generative format using our proposed instruction template. EvalQA, on the other hand, involves tasking the MLLM to predict the correctness of a given visual-question-answer triplet. Recognizing the absence of datasets specifically designed to assess VQA correctness, we have developed a new benchmark called **EvalQABench** for evaluating VQA data. Rather than asking humans to label such a dataset, we propose a new pipeline for data construction. This benchmark comprises training, validation, and test sets, with each VQA pair accompanied by a "Yes" or "No" label indicating correctness, along with a one-sentence explanation as the feedback. For instance, *"Yes, the oranges are not in a bag"*.

By integrating the GenQA and EvalQA tasks into the vanilla multimodal learning, we develop an effective training framework called **LOVA³**. In this study, we select the SOTA MLLM LLaVA-1.5 as the backbone model for evaluation. We conduct experiments on 10 widely used multimodal benchmarks such as GQA [30], VQAv2 [26], Vizwiz [27], MME [19], MMBench [45], and MM-vet [92], and observe consistent improvements across these benchmarks. To summarize, our proposed LOVA³ is a new framework that endows the MLLM with the ability to ask and assess and finally achieve profound multimodal understanding capability. Overall, our contributions are three folds:

(1) To the best of our knowledge, **LOVA³** is the first effort to imbue the asking and assessment abilities in training a robust and intelligent MLLM. LOVA³ open an avenue for imitating the human abilities towards holistic intelligence for MLLM.

(2) We build a new benchmark **EvalQABench** for the VQA evaluation as the first effort to advance the VQA data assessment of future research.

(3) The experimental results demonstrate that training with LOVA³ consistently improves performance across several multimodal benchmarks, including VQAv2, GQA, MME, VizWiz, MMBench, and MM-Vet, etc.

## 2 Related Work

### 2.1 Multimodal Large Language Models

Large Language Models (LLMs) [16, 98, 65, 5, 81, 15, 82] such as GPT-4 [61] demonstrate their exceptional capacity to handle a wide range of complex tasks to play an important role in assisting humans in daily life. Equipped with these LLMs, a surge of multimodal modes [37, 43, 17, 3, 10, 84, 101, 9, 8, 51, 85, 29, 62, 58, 88, 22, 47, 28, 34, 7, 100, 48, 14, 38, 24, 68, 23, 66] are proposed to integrate the visual information with the pre-trained LLM decoder for diverse multimodal reasoning tasks such as image captioning [12, 1, 91] and visual question answering [26, 54, 30, 27]. LLaVA [43, 42] is a pioneering approach that collects 665K instruction tuning data from present vision-language (VL) datasets for supervised finetuning (SFT) and achieves promising results on various datasets in a lower cost of training requirement. Another SOTA model InstructBLIP [17] also proposes gathering datasets to construct their instruction tuning dataset. It adopts the VQG task but is limited in generic data type. Different from InstructBLIP, we propose the GenQA task for jointly generated questions and answers on 5 primary VL tasks not restricted to generic data type. Besides focusing on traditional vision-language tasks, Shikra [10], Kosmos-2 [62], PVIT [8], Ferret [90] pay attention to the image-region based multimodal tasks (i.e., Referring Expression Comprehension) and demonstrate the performance improvement with these hard tasks. By adopting a large-scale image-text corpus for instruction tuning, Qwen-VL [3], CogVLM [84], AnyMAL [58] and Chameleon [79] achieve exceptional performance on various multimodal tasks. However, these MLLMs primarily concentrate on training the model to answer questions as effectively as possible, neglecting the significance of enabling the model to act as a questioner and a competent evaluator within the training paradigm.

### 2.2 Visual Question Answering and Generation

Nine years ago, visual question answer [2] was defined and became an essential task for evaluating multimodal systems. A surge of VQA-related benchmarks [2, 26, 76, 56, 55, 27, 71, 54, 50, 30, 39] are emerged to advance the development of this research area, including generic VQA benchmarks [2, 26, 30], text-based VQA [76, 56, 55, 78], knowledge-augmented VQA [54, 71, 50, 13], and goal-oriented VQA [27] aimed at assisting blind people.

Besides the VQA task, the Visual Question Generation (VQG) task was first formulated in [60], which contributes a VQG dataset with each image annotated with multi-questions and benchmarking on generative models and retrieval models. [97] first employs an RNN-based encoder-decoder framework alongside model-generated captions to generate questions. After that a list of works [59, 60, 18, 32, 72, 87, 83] are proposed for promoting this research area. Two interesting studies [40, 73] pose that treating VQG as a complementary task can enhance the robustness of visual question answering. This finding reaffirms our motivation that training a model to generate diverse questions contributes to a deeper understanding of visual information, thereby improving its problem-solving capabilities. Unlike the traditional Visual Question Generation (VQG) task, which focuses primarily on the generic VQA domain, GenQA is designed to generate diverse VQA data including MC VQA, MT, REC, and REG. Additionally, GenQA generates both questions and answers simultaneously, whereas traditional VQG focuses solely on question generation.

### 2.3 Multimodal Benchmarks

Traditional multimodal benchmarks focus on answering ability, such as visual question answering [26], image captioning [12, 63, 1], as well as other benchmarks for specialized scenarios such as scene text understanding [76, 75], commonsense reasoning [94], outside knowledge [54, 71]. The recent development of MLLM posts a strong need for modernized multimodal benchmarks [19, 45, 36, 92, 25, 93, 49, 20, 11, 96, 44, 99, 77, 80] such as MME [19], MMBench [45], SEED-Bench [36] which involve comprehensively evaluating current MLLMs on various multimodal abilities. Unlike existing multimodal benchmarks focusing primarily on evaluating the model's ability to answer, we introduce EvalQABench, a benchmark designed to evaluate the correctness of VQA pairs, each with a binary "Yes/No" annotation. Furthermore, recognizing the lack of emphasis on providing feedback for incorrect answers in current benchmarks, we develop an LLM-based pipeline. This pipeline can automatically generate feedback, paving the way for enhanced automated data processing in the future.

Table 1: Data taxonomy of GenQA, detailing the data type, name, size, and instruction prompts of each dataset.

| Data Type | Dataset | Size | Instruction Prompts |
|---|---|---|---|
| Generic VQA | VQAv2 [26] | 100K | *Note: randomly choose from 58 instruction prompts* |
| | GQA [30] | 100K | Example: Can you provide a clear question and its answer based on the image? |
| | OCR-VQA [56] | 80K | |
| | Counting20K[†] | 20K | |
| | LLaVA-250K[†] [43] | 250K | |
| Multi-choice VQA | A-OKVQA [71] | 17K | Can you provide a clear question and its answer based on the image?\nThis is a Multi-choice VQA task. |
| Multi-turn VQA | VQAv2 [26] | 83K | Design a conversation between you and a person asking about this photo. |
| | GQA [30] | 72K | The answers should be in a tone that a visual AI assistant is seeing the image and answering the question. Ask diverse questions and give corresponding answers. |
| REC | VG [33] | 30K | *Note: randomly choose from 58 instruction prompts with a specific task description prompt.* Can you review the image and articulate a concise question and its answer?\nThis is a Referring Expression Comprehension (REC) task. The question will express a specific region of the image. Please provide the coordinates in the answer. |
| | RefCOCO [31] | 30K | |
| REG | VG [33] | 30K | *Note: randomly choose from 58 instruction prompts with a specific task description prompt.* Can you review the image and articulate a concise question and its answer?\nThis is a Referring Expression Generation (REG) task. The purpose of REG is to generate a unique description for a specified location. |
| | RefCOCO [31] | 30K | |
| Total | - | 842K | |

# 3 Methodology

In this section, we introduce LOVA[3], a new framework designed to imitate two essential abilities - asking and assessment - within multimodal learning. We delve into the specifics of addressing this challenge through GenQA data collection, EvalQA data creation, model architecture, and training.

## 3.1 Data Collection for GenQA

If one MLLM is able to successfully generate high-quality question-answer pairs based on visual input, it indicates a stronger problem-solving ability and deep visual understanding [40, 73]. To enable the MLLM to ask questions, it is natural for us to gather existing annotated datasets as the training corpus and then train the model to predict both questions and answers. We carefully define five main multimodal data types as listed in Tab. 1. For each data type, we gather widely used human-annotated datasets or high-quality instruction tuning datasets generated by GPT-4. We select Generic VQA tasks to generate fundamental questions, e.g., object count and object action. We incorporate Multi-choice VQA (MC VQA) and Multi-turn VQA (MT) to increase the diversity of data formats. Additionally, we include two multimodal grounding tasks: Referring Expression Comprehension (REC) and Referring Expression Generation (REG). Generating REC and REG data requires a deeper understanding of image content, enabling the model to fully comprehend visual cues. Both tasks increase the difficulty of GenQA, which helps MLLM acquire a higher level of multimodal understanding. In total, we gather 842K data for training questioning ability.

## 3.2 Data Creation for EvalQA

Completing the VQA assessment often requires fine-grained and deep visual understanding. As emphasized in Sec. 1, the ability to assess is often overlooked yet crucial in MLLM training. To address this gap, we introduce a new benchmark, **EvalQABench**, to address the problem of assessing visual question-answering data. Moreover, instead of merely labeling each VQA pair with "Yes/No", we advocate for integrating **feedback** into each instance, an important aspect rarely seen in prior multimodal benchmarks. We consider training the model not only to assess the correctness of the answer but also to provide reasonable feedback that would increase the capability for multimodal understanding. EvalQABench comprises three datasets: training, validation, and test sets. As illustrated in Tab. 2, we present examples of the training set from EvalQABench across various question types.

**MLLM-based Negative Answer Generation.** The main challenge of EvalQABench lies in constructing negative answers. When dealing with large-scale ground-truth VQA pairs, how can we automatically produce the negative answer? One viable solution is to leverage a multimodal model for this purpose. Recognizing that Fuyu-8B [4] is an open-source free MLLM that stands out with the exceptional ability to process high-resolution images and perform robust well on many complex tasks. We utilize it to generate negative answers with the following prompt:

Table 2: Selected examples from **EvalQABench** training set, including the ground truth answer, negative answer, and feedback.

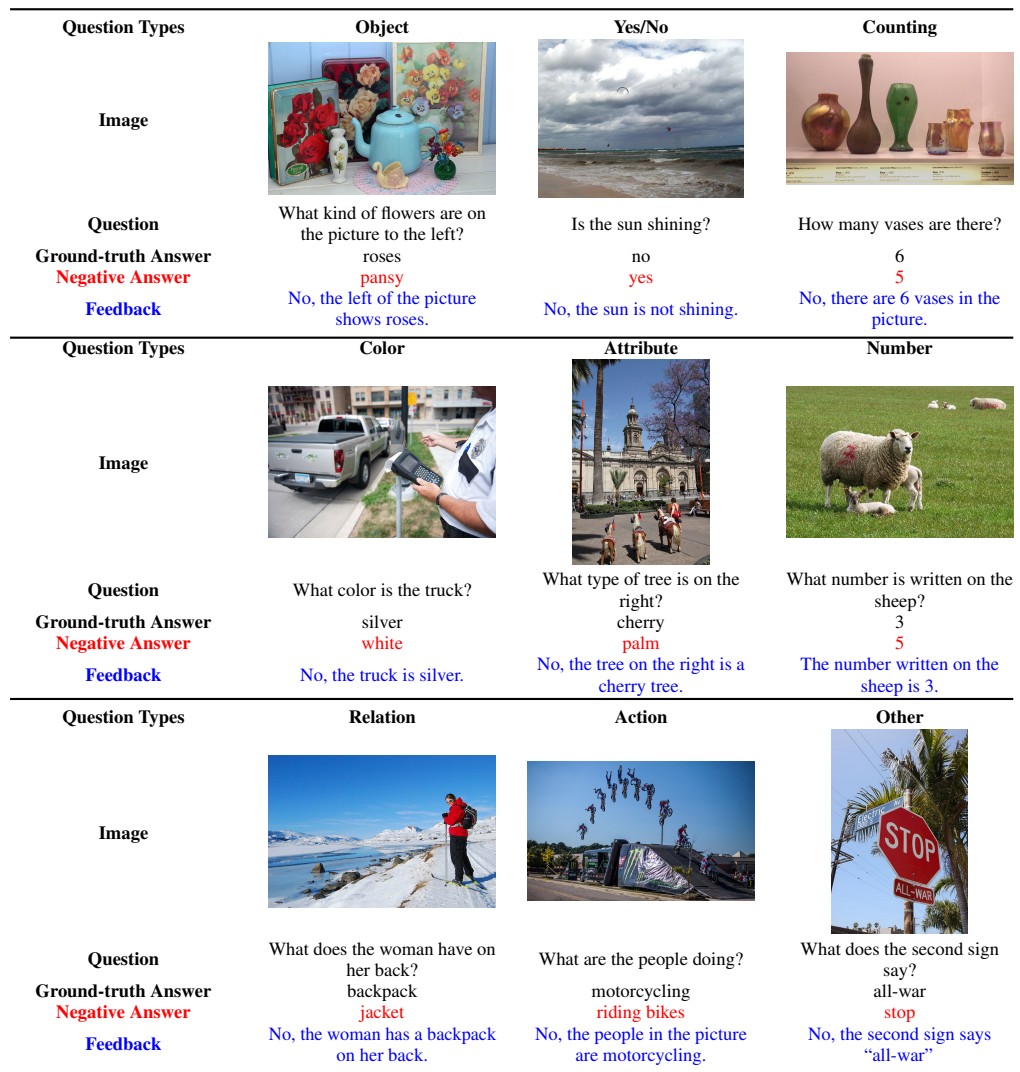

| Question Types | Object | Yes/No | Counting |
|---|---|---|---|
| Image | | | |
| Question | What kind of flowers are on the picture to the left? | Is the sun shining? | How many vases are there? |
| Ground-truth Answer | roses | no | 6 |
| Negative Answer | pansy | yes | 5 |
| Feedback | No, the left of the picture shows roses. | No, the sun is not shining. | No, there are 6 vases in the picture. |
| Question Types | Color | Attribute | Number |
| Image | | | |
| Question | What color is the truck? | What type of tree is on the right? | What number is written on the sheep? |
| Ground-truth Answer | silver | cherry | 3 |
| Negative Answer | white | palm | 5 |
| Feedback | No, the truck is silver. | No, the tree on the right is a cherry tree. | The number written on the sheep is 3. |
| Question Types | Relation | Action | Other |
| Image | | | |
| Question | What does the woman have on her back? | What are the people doing? | What does the second sign say? |
| Ground-truth Answer | backpack | motorcycling | all-war |
| Negative Answer | jacket | riding bikes | stop |
| Feedback | No, the woman has a backpack on her back. | No, the people in the picture are motorcycling. | No, the second sign says "all-war" |

> `` This is the question: `<Q>` . Please give me the wrong answer to this question. The answer should be a single word or phrase.\n

Here, `` and `<Q>` are two placeholders for the image and question from ground truth VQA pair. The output of Fuyu-8B provides a negative answer, such as *"pansy"*, as illustrated in Fig. 2.

**Manual Filtering and Error Correction.** Acknowledging that the Fuyu-8B model is not flawless and recognizing that no multimodal model, including GPT-4V, is perfect, we have implemented both manual filtering and error corrections, as illustrated in Fig. 3 for the post-data processing. Through empirical analysis, we identified 4 primary types of errors. For instance, an answer generated by Fuyu-8B may be present in the question but lacks semantic relevance or is identical to the correct answer. Additionally, some incorrect answers may result from misunderstanding the question's category, as exemplified by the example in Fig. 3. Beyond filtering, we propose error corrections for two types of questions: "Yes/No" and "Counting". For "Yes/No" questions, we directly substitute an incorrect answer with "Yes". For "Counting" questions, we first verify if the English numeral matches the correct answer; if not, we replace it with a random number. After applying the above filtering and correction processes, we found that most of the incorrect samples had been removed.

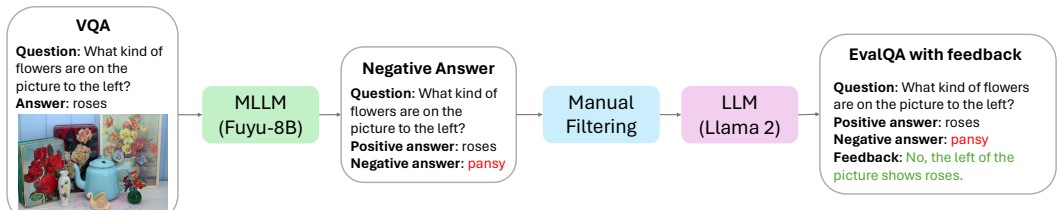

Figure 2: Illustration of the proposed pipeline for generating negative answers and feedback.

**LLM-based Feedback Generation.** With the candidate's negative answer, we then focus on generating error feedback. We consider the feedback describing the reason for incorrectness will help the MLLM obtain a deeper understanding. We thus utilize the LLM Llama 2 [82] to generate the feedback by reasoning the ground truth question-answer pairs with the following prompt:

> Please rephrase the question and answer: `<Q> \n <A>` into one short description.

After processing by Llama 2, we can get feedback like *"No, the left of the picture shows roses."*. Moreover, we use similar manual filtering strategies that are used in the negative answer generation step to remove the noisy samples with wrong formats or empty output.

In summary, we start by randomly selecting 100,000 samples from 443,758 annotated VQA pairs in the VQAv2 training set [26] to generate negative answers. After manual filtering, this number is reduced to 61,094 samples. We then generate feedback for each sample and further filter out those with incorrect formats, resulting in a final set of 41,592 samples. For the training set of EvalQABench, we create a one-positive-one-negative format by randomly selecting 32,000 negative samples from the 41,592 filtered samples, yielding a total of 64,000 training data points. For the validation and test subsets, we follow a similar sampling procedure. We randomly select 100,000 samples from the VQAv2 validation set, resulting in 41,907 negative samples. From these, we randomly select 2,500 negative samples each for the validation set and the test set.

## 3.3 Model Architecture

In this subsection, we introduce the model architecture of LOVA$^3$. This model is built upon the prevalent MLLM LLaVA-1.5 [42] with three key components: Vision Encoder, MLP Adapter, and Large Language Model. For the vision encoder, we follow LLaVA-1.5 [42] and implement it with a pre-trained CLIP-Large vision encoder [64] with resolution $336 \times 336$. For the large language model, we adopt the widely used instruction fine-tuned model, Vicuna-7B [15]. Following [42], the MLP adapter is a simple two-layer MLP since such a simple design is better for reserving the visual information while achieving running efficiency. In this study, we leverage LLaVA-1.5 to build upon because of its exceptional performance and highly reproducible training and validating codes. Other outstanding MLLMs, such as CogVLM [84] and Qwen-VL [3], are pre-trained on billions-scale datasets or in-house datasets. This scale of data makes the training process difficult to replicate and poses challenges in incorporating our proposed training tasks, GenQA and EvalQA.

## 3.4 Training

For brevity, we denote the LOVA$^3$ model as $F_M$. Given an image $X_I$, our target is to enforce $F_M$ to generate the response $X_R$:

$$X_R = F_M(X_T, X_I), \tag{1}$$

where $X_T$ represents the input text. $X_T$ can be an example of the three types: 1) VQA data, e.g., *"What color is the pot?"*; 2) GenQA data like *"Can you provide a concise question and answer based on the image?"*; 3) EvalQA data, such as *"What kind of flowers are on the picture to the left?\nAnswer: pansy. \nPlease examine the correctness of this question and answer according to the image content. Output Yes or No with the feedback"*. Accordingly, the input instruction template can be unified into the following ones:

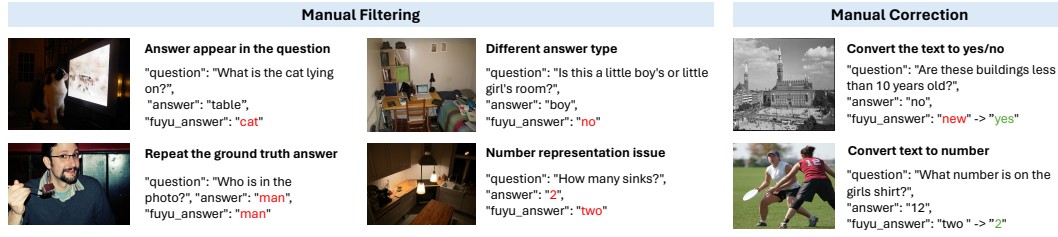

Figure 3: Examples from the manual filtering and error correction process. Red text indicates error answers, while Green text represents manually corrected answers.

USER: $X_I$ $X_T$ \n ASSISTANT: $X_R$ 

We follow previous MLLMs[42, 17], and design the training objective in an autoregressive manner:

$$\max \sum_{i=i}^{L} \log p(X_R|X_T, X_I) = \prod_{i}^{L} p_\theta(x_i|X_T, X_I, X_{R,<i}),  \qquad (2)$$

where $x_i$ is the current prediction token and $L$ denotes the response sequence length. $\theta$ denotes the trainable parameters.

## 4 Experiments

### 4.1 Datasets and Settings

**Training Datasets.** For the fair comparison, we utilize the 665K instruction-following dataset introduced in LLaVA1.5, combined with the 842K GenQA data as outlined in Tab. 1, and an additional 64K data comprising one-positive-one-negative pairs as described in Section 3.2, totaling our training datasets with 1.5M samples. It is important to note that the datasets and annotations used in both VQA and GenQA are the same. There are no additional datasets involved, thus avoiding unfair comparisons caused by the introduction of new instruction data. For EvalQA, we adopt VQAv2 to build the training set, which is already included in the original 665K instruction dataset.

**Validation Datasets.** We assess LOVA[3] on 10 widely used multimodal datasets and benchmarks. (1) VQAv2 [26] and GQA [30] are two large-scale annotated VQA datasets comprising 430K and 943K instances. (2) VizWiz [27] is a challenging dataset comprising 8000 instances of test-dev set. Most of the images in this dataset are blurred, making it difficult to respond. (3) ScienceQA [50] is a benchmark comprising 21k multimodal multiple-choice questions with diverse science topics. (4) POPE [39] is a benchmark for evaluating the object hallucination in the MLLM. (5) MME [19], SEED-Bench [36], MMBench [45], LLaVA-Bench [43], MM-Vet [92] are five prominent multimodal benchmarks designed to evaluate various capabilities of MLLMs, including object existence, color recognition, counting, OCR, etc.

**Competitors.** We compare LOVA[3] with other SOTA models inlcuding MiniGPT-4 [101], BLIP2 [37], InstructBLIP [17], mPLUG-owl [89], LLaMA-AdapterV2 [22] and LLaVA-1.5 [42]. We report the results from their paper or the benchmark leaderboard.

**Implementation Details.** To ensure a fair comparison, we train the LOVA[3]-7B model without tuning any hyperparameters of LLaVA-1.5 [42] from its original supervised finetuing stage. The model is trained for one epoch across three tasks: VQA, GenQA, and EvalQA. Specifically, we employ the AdamW [46] optimizer with a learning rate of $2 \times 10^{-5}$ and a total batch size of 128. The training process takes 24.5 hours on an 8 Nvidia A100 (40G) GPU setup. Moreover, we also replace the LLM from Vicuna-7B[15] to Phi-1.5B[41] to evaluate smaller LLMs. We train LLaVA-Phi-1.5 and LOVA[3]-1.5B by using the same training recipe. The only difference of training with Phi-1.5 is that we increase the learning rate from $2 \times 10^{-5}$ to $4 \times 10^{-5}$ to ensure the higher performance. The model LLaVA-Phi-1.5 is trained with the original 665K VQA instruction data as the baseline. The model LOVA[3]-1.5B is trained with our proposed 1.5M mixture data including VQA, GenQA, EvalQA data.

Table 3: Results on five generic tasks including VQAv2 [26], GQA [30], VizWiz [27], ScienceQA [50], and POPE [39]. The first two columns represent the results on held-in datasets marked as $^*$, and the last three columns represent the held-out datasets. The best result on each subtask is **bolded**.

| Method | Train Paradigm | LLM | VQAv2 test-dev | GQA test | VizWiz test-dev | ScienceQA img | POPE avg |
|--------|---------------|-----|--------|-----|---------|-----------|------|
| LLaVA-Phi-1.5[42] | VQA | Phi-1.5B | 73.2$^*$ | 56.1$^*$ | 33.8 | 57.3 | **87.6** |
| LOVA$^3$-1.5B (ours) | VQA, GenQA, EvalQA | Phi-1.5B | **75.8**$^*_{+2.6}$ | **58.6**$^*_{+2.5}$ | **37.2**$^*_{+3.4}$ | **57.8**$^*_{+0.5}$ | 86.0$^*_{-1.6}$ |
| BLIP-2 [37] | VQA | Vicuna-13B | 41.0 | 41.3 | 19.6 | 61.0 | 85.3 |
| InstructBLIP [17] | VQA, VQG | Vicuna-7B | – | 49.2 | 34.5 | 60.5 | – |
| InstructBLIP [17] | VQA, VQG | Vicuna-13B | – | 49.5 | 33.4 | 63.1 | 78.9 |
| IDEFICS-9B [35] | VQA | LlamA-7B | 50.9 | 38.4 | 35.5 | 44.2 | – |
| Qwen-VL [3] | VQA | Qwen-7B | 78.8$^*$ | 59.3$^*$ | 35.2 | 67.1 | – |
| LLaVA-1.5 [42] | VQA | Vicuna-7B | 78.5$^*$ | 62.0$^*$ | 50.0 | 66.8 | 85.9 |
| LOVA$^3$-7B(ours) | VQA, GenQA, EvalQA | Vicuna-7B | **80.3**$^*_{+1.8}$ | **63.3**$^*_{+1.3}$ | **53.6**$_{+3.6}$ | **68.0**$_{+1.2}$ | **87.4**$_{+1.5}$ |

Table 4: Results on multimodal benchmarks, including MME [19] and SEED-Bench [36], MM-Bench [45] and LLava-Bench [43]

| Method | Train Paradigm | LLM | MME | SEED-Bench Image | MMBench En | MMBench Cn | LLaVA-Bench All |
|--------|---------------|-----|-----|------------------|----|----|-----|
| LLaVA-Phi-1.5 [42] | VQA | Phi-1.5B | 1114.7 | 58.2 | 53.7 | 4.1 | 59.0 |
| LOVA$^3$-1.5B (ours) | VQA, GenQA, EvalQA | Phi-1.5B | **1212.9**$_{+98.2}$ | **60.1**$_{+1.9}$ | **55.9**$_{+2.2}$ | **10.4**$_{+6.3}$ | **59.1**$_{+0.1}$ |
| BLIP-2 [37] | VQA | Vicuna-13B | 1293.8 | 49.7 | – | – | 38.1 |
| InstructBLIP [17] | VQA, VQG | Vicuna-7B | – | 58.8 | 36.0 | 23.7 | 60.9 |
| InstructBLIP [17] | VQA, VQG | Vicuna-13B | 1212.8 | – | – | – | 58.2 |
| mPLUG-owl [89] | VQA | Llama-7B | 967.3 | 37.9 | – | – | – |
| LLaMA-AdapterV2 [22] | VQA | Llama-7B | 972.7 | 35.2 | 41.0 | – | – |
| LLaVA-1.5 [42] | VQA | Vicuna-7B | 1510.7 | 66.2 | 64.3 | 58.3 | 64.0 |
| LOVA$^3$-7B (ours) | VQA, GenQA, EvalQA | Vicuna-7B | **1552.7**$_{+42.0}$ | **67.1**$_{+0.9}$ | **66.8**$_{+2.5}$ | **60.5**$_{+2.2}$ | **68.3**$_{+4.3}$ |

## 4.2 Main Results

**Generic tasks.** As shown in Tab. 3, LOVA$^3$-7B outperforms LLaVA1.5 across all five datasets and obtains 3.6% improvement on VizWiz dataset, 1.3% improvement on GQA, 1.8% improvement on VQAv2 (1,932 samples are correctly predicted), and 1.2% improvement on ScienceQA. As for the object hallucination benchmarks, our model attains 87.4% accuracy at an average of its three subsets. Remarkably, these enhancements in VQAv2 and GQA performance are achieved without any extra datasets, underscoring the significant impact of integrating GenQA and EvalQA into our training to promote performance improvements on these generic VQA tasks. Based on the results from smaller LLMs, our LOVA$^3$-1.5B outperforms the baseline LLaVA-Phi-1.5 on VQAv2, GQA, VizWiz, and ScienceQA by 2.6%, 2.5%, 3.4%, and 0.5%, respectively. This demonstrates a consistent improvement when training with our LOVA$^3$ framework across varying LLM sizes. Additionally, a comparison of improvements for both 7B and Phi-15B on the VizWiz dataset highlights the advantage of our LOVA$^3$ framework for this VQA task.

**MME, SEED-Bench, MMBench, LLaVA-Bench.** In Tab. 4, we evaluate four prevalent multimodal benchmarks, where our LOVA$^3$-7B surpasses LLaVA1.5 with 42.0% on MME benchmark, 0.9% increase in accuracy on SEED-Bench, 2.5% on MMBench (En), 2.2% MMBench (Cn) and 4.3% on LLaVA-Bench. Such results showcase enhanced multimodal reasoning capabilities for complex tasks compared to vanilla LLaVA1.5, which is solely trained with VQA tasks. By investigating the results produced by LOVA$^3$-1.5B on these multimodal benchmarks, one can see greater improvements in the smaller models. Notably, LOVA$^3$-1.5B achieves an impressive 98.2% improvement on the MME benchmark. The lower results of LLaVA-Phi-1.5 and LOVA$^3$-1.5B on MMBench (Cn) may be attributed to a lack of Chinese training data in the Phi-1.5 training process.

**MM-Vet.** In Tab. 5, we compare LOVA$^3$-7B with other approaches on MM-Vet, which is a challenging benchmark including numerous complex VQA samples that demand integration of several multimodal capabilities for answering. As illustrated in Tab. 5, the results show that our LOVA$^3$-7B outperforms LLaVA-1.5 by 4.0% at an average. Such improvement demonstrates the effectiveness of LOVA$^3$-7B in solving these challenging multimodal questions. Based on the results on LOVA$^3$-1.5B and LLaVA-Phi-1.5, one can see a greater improvement than LOVA$^3$-7B.

Table 5: Multimodal reasoning ability on MM-Vet [92]. Rec denotes Recognition; Know denotes knowledge; Gen denotes Language generation; and Spat denotes Spatial awareness.

| Method | Train Paradigm | LLM | Rec | OCR | Know | Gen | Spat | Total |
|---|---|---|---|---|---|---|---|---|
| LLaVA-Phi-1.5 [42] | *VQA* | Phi-1.5B | – | – | – | – | – | 22.2 |
| LOVA$^3$-1.5B (ours) | *VQA, GenQA, EvalQA* | Phi-1.5B | – | – | – | – | – | **28.1**$_{+5.9}$ |
| MiniGPT-4 [101] | *VQA* | Vicuna-7B | 27.4 | 15.0 | 12.8 | 13.9 | 20.3 | 22.1 |
| BLIP-2 [37] | *VQA* | Vicuna-13B | 27.5 | 11.1 | 11.8 | 7.0 | 16.2 | 22.1 |
| InstructBLIP [17] | *VQA, VQG* | Vicuna-7B | 32.4 | 14.6 | 16.5 | 18.2 | 18.6 | 26.2 |
| InstructBLIP [17] | *VQA, VQG* | Vicuna-13B | 30.8 | 16.0 | 9.8 | 9.0 | 21.1 | 25.6 |
| LLaVA-1.5 [42] | *VQA* | Vicuna-7B | 37.0 | 21.0 | 17.6 | 20.4 | 24.9 | 31.2 |
| LOVA$^3$-7B (ours) | *VQA, GenQA, EvalQA* | Vicuna-7B | **41.5**$_{+4.5}$ | **23.6**$_{+2.6}$ | **23.9**$_{+6.3}$ | **24.6**$_{+4.2}$ | **30.3**$_{+5.4}$ | **35.2**$_{+4.0}$ |

Table 6: Abaltion studies on different finetuning datasets. The model is LOVA$^3$-7B.

| Row | Finetuning Corpus | | | GQA | VizWiz | ScienceQA | POPE | MME | Size |
|---|---|---|---|---|---|---|---|---|---|
| | GenQA-Generic | GenQA-Grounding | EvalQA | | | | | | |
| 0 | | LLaVA-1.5 (Baseline) | | 62.0 | 50.0 | 66.8 | 85.9 | 1510.7 | 665K |
| 1 | ✓ | | | 63.1 | 53.1 | 67.4 | 86.9 | 1550.7 | 722K |
| 2 | | ✓ | | 62.8 | 50.9 | 66.4 | 86.6 | 1495.8 | 120K |
| 3 | | | ✓ | 62.8 | 49.1 | 67.8 | 87.0 | 1535.6 | 64K |
| 4 | ✓ | ✓ | | 63.3 | 53.2 | 67.4 | 86.7 | 1523.6 | 842K |
| 5 | ✓ | | ✓ | **63.7** | **54.4** | 67.0 | 86.9 | 1520.8 | 786K |
| 6 | | ✓ | ✓ | 63.1 | 51.1 | 67.5 | 86.8 | 1478.7 | 184K |
| 7 | ✓ | ✓ | ✓ | 63.3 | 53.6 | **68.0** | **87.4** | **1552.7** | 906K |

Table 7: Results on 10 multimodal datasets. The 64K training data for EvalQA task is generated by the Gemini-1.5-Flash model.

| Method | LLM | VQAv2 test-dev | GQA test | VizWiz test-dev | ScienceQA img | POPE avg | MME | SEED Image | MMBench En | Cn | LLaVA All | MM-Vet Total |
|---|---|---|---|---|---|---|---|---|---|---|---|---|
| LLaVA-Phi-1.5 [42] | Phi-1.5B | 73.2* | 56.1* | 33.8 | 57.3 | **87.6** | 1114.7 | 58.2 | 53.7 | 4.1 | 59.0 | 22.2 |
| LOVA$^3$-1.5B (ours) | Phi-1.5B | **75.8*** | **58.4*** | **36.9** | **57.8** | 85.8 | **1202.9** | **60.5** | **55.5** | **7.82** | **60.0** | **25.1** |
| LLaVA-1.5 [42] | Vicuna-7B | 78.5* | 62.0* | 50.0 | 66.8 | **85.9** | 1510.7 | 66.2 | 64.3 | **58.3** | 64.0 | 31.2 |
| LOVA$^3$-7B (ours) | Vicuna-7B | **80.3*** | **63.4*** | **54.2** | **70.8** | 85.6 | **1526.8** | **67.6** | **66.5** | 57.6 | **67.7** | **32.2** |

## 4.3 Ablation Study

We split the data used in the GenQA task into two groups: GenQA-General and GenQA-Grounding. The findings, presented in Tab. 6, are instrumental in investigating the contributions of GenQA and EvalQA to model efficacy. **(1)** Comparing the first four rows, one can find that both GenQA-General and EvalQA data are more effective in improving performance than GenQA-Grounding. **(2)** By comparing rows 4 and 7, it demonstrates the effectiveness of EvalQA across five datasets, especially on MME. **(3)** When comparing rows 6 and 7, by removing GenQA-General from the finetuning corpus, the performance drops significantly on MME and VizWiz. **(4)** Compare the rows 0 and 3, one can observe that even adding 64K data into the training, there are obvious improvements in GQA, ScienceQA, and MME. By analyzing the data size, we did not introduce any new datasets for training the GenQA task. For EvalQA, we only added 32K new negative answer annotations while retaining the original questions used for training VQA capabilities. The details of the data size are provided in the right column in Tab. 6.

## 4.4 Training with Gemini-Generated EvalQA Data

Rather than using the open-source model Fuyu-8B [4] to create the training data for EvalQABench, we also explore the use of the commercial model Gemini-1.5-Flash[1] as both the MLLM and LLM in Fig. 2 to generate negative answers and one-sentence feedback. The experimental results, presented in Tab. 7, indicate that regardless of whether we use the open-source model Fuyu-8B or the commercial model Gemini-1.5-Flash, our proposed training paradigm LOVA$^3$ consistently improves performance across both smaller and larger baseline models.

---

[1]https://deepmind.google/technologies/gemini/flash/

Table 8: Results of multimodal large language models on the test set of **EvalQABench (ours)**.

| Method | LLM | Test Set | | | No (%) |
|--------|-----|----------|-----------|----------|--------|
| | | Accuracy | Precision | F1 Score | |
| *Vision Language Pretraining Model* | | | | | |
| BLIP2 [37] | Flan-T5-XXL-11B | 58.00 | 82.79 | 32.47 | 87.80 |
| *Multimodal Large Language Models* | | | | | |
| InstructBLIP [17] | Vicuna-7B | 38.04 | 41.49 | 48.47 | 29.76 |
| InstructBLIP [17] | Vicuna-13B | 61.42 | 57.60 | 69.18 | 24.82 |
| CogVLM [84] | Vicuna-7B | 60.64 | 56.59 | 69.88 | 19.32 |
| Qwen-VL-Chat [3] | Qwen-7B | 63.66 | 63.48 | 63.90 | 49.34 |
| InternLM-XC [95] | InternLM-7B | 69.58 | 70.66 | 68.76 | 52.62 |
| LLaVA-1.5 [42] | Vicuna-7B | 64.92 | 61.28 | 69.80 | 33.84 |
| LOVA$^3$-7B (ours) | Vicuna-7B | $\mathbf{79.58}_{+14.66}$ | $\mathbf{79.15}_{+17.87}$ | $\mathbf{79.72}_{+9.92}$ | 49.26 |

## 4.5 Benchmark of EvalQABench

We report the evaluation results on our EvalQABench test set in Tab. 8 to validate the EvalQA ability of current SOTA models and LOVA$^3$. We select BLIP2 [37], InstructBLIP [17], CogVLM [84], Qwen-VL-Chat [3], InternLM-XC [95], and LLaVA1.5 [42] for the comparison. We ask these models to answer "Yes" or "No" strictly and record the results for calculating the Accuracy, Precision, F1 score, and No (%) metrics. Here, No (%) indicates the percentage of results classified as "No," which ideally should approximate 50% due to the one-positive-one-negative setting utilized in our test set. As indicated by the data presented in the table, BLIP2 predominantly yields "No" responses across most test instances. Among the state-of-the-art MLLMs, InternLM-XC stands out by delivering superior performance on these four metrics. Trained with EvalQA data, LOVA$^3$ shows several improvements over our baseline LLaVA1.5 by margins of 14.66%, 17.87%, and 9.92% in Accuracy, Precision, and F1 Score, respectively.

## 5 Conclusion and Limitations

In this work, we propose a novel multimodal framework, **LOVA**$^3$, which is capable of mimicking the human visual question answering, asking, and assessment to achieve deeper multimodal understanding. We introduce two additional training tasks, **GenQA** and **EvalQA**, to help MLLM acquire these abilities. We establish **EvalQABench**, a novel benchmark to assess the VQA samples between multiple MLLMs. Experimental results show that LOVA$^3$ achieves superior performance across various benchmarks, including MM-Vet, SEED, and VizWiz, demonstrating the effectiveness of the two additional abilities.

**Limitations.** (1) Due to computational constraints, we do not test larger LLMs, such as the 13B or 34B variants. However, we believe that our LOVA$^3$ could be beneficial for larger LLMs, as other MLLMs have shown performance improvements with increased LLM scale. (2) GenQA and EvalQA as two additional tasks increase training costs, but it is inevitable for an MLLM to acquire new capabilities. (3) Due to the limited scope of instruction tuning datasets, LOVA$^3$ cannot address domain-specific multimodal tasks well, such as text-centric VQA or mathematic-relevant VQA.

## 6 Acknowledgement

This research is supported by National Research Foundation, Singapore and A*STAR, under its RIE2020 Industry Alignment Fund – Industry Collaboration Projects (IAF-ICP) grant call (Grant No. I2001E0059) – SIA-NUS Digital Aviation Corp Lab. Mike Zheng Shou is supported by the National Research Foundation, Singapore under its NRFF Award NRF-NRFF13-2021-0008. Pan Zhou was supported by the Singapore Ministry of Education (MOE) Academic Research Fund (AcRF) Tier 1 grants (project ID: 23-SIS-SMU-028 and 23-SIS-SMU-070).

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

## A    Broader Impacts

In this paper, we propose a new training framework for imbuing two essential abilities into the model training. We also propose the EvalQA task with a new benchmark of 64,000 training data and 5,000 validation and testing data. In summary, this work would inspire future work to pay more attention to visual question asking and assessment. For these two tasks, there still exists some space for involving more GenQA tasks and more formulations of EvalQA.

## B    Additional Details of Implementation

**665K instruction tuning data.** We follow the backbone MLLM LLaVA1.5 to adopt the 665K instruction-following data into the supervise finetuning stage. We present the details of the 665K instruction data in Tab. 9 for convenient browsing. The details include the dataset name, size, and instruction prompts.

**Model training.** The baseline model LLaVA1.5 [42] includes two stages: image-text alignment pertaining stage and supervised instruction tuning stage. The first stage involves only the image caption datasets for aligning two modalities by only finetuning the two-layer MLP adapter. In this paper, we are investigating the effectiveness of two additional tasks in the supervised instruction tuning stage. Thus, we use the pretraining weights of the first stage for fair comparison and train the model as in Fig. 4. The model comprises three key components: Vision Encoder, MLP Adapter, and Large Language Model. The vision encoder is responsible for processing the input image $X_I$ to align the learned visual features with the text input $X_T$. Next, the Multi-Layer Perceptron (MLP) adapter projects the visual feature $F_I$ to $Z_I$. Finally, the large language model utilizes $Z_I$ and the current text embedding $Z_T$ to predict the response in a left-to-right manner. During training, the data of the three tasks is mixed. By such joint training, MLLM exhibits deeper comprehension and promising reasoning ability.

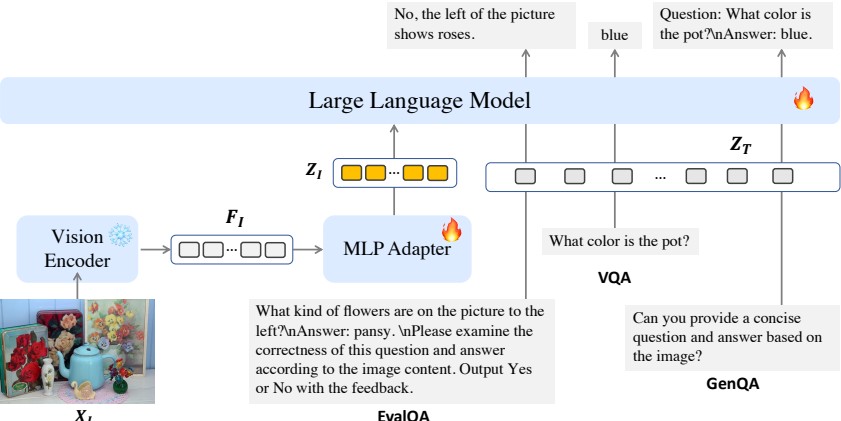

Figure 4: Illustration of the model training of LOVA$^3$.

**Hyperparameter.** The hyperparameters of LOVA$^3$ are aligned with those of LLaVA1.5 to ensure a fair comparison, as illustrated in Tab 10. The exceptional performance highlighted in Tab. 3, 4, and 5 of the main paper, achieved without any modulation of hyperparameters, demonstrates the effectiveness and robustness of our LOVA$^3$.

## C    Details of GenQA Data

**Generic VQA.** It includes four datasets: VQAv2 [26], GQA [30], OCR-VQA [56] and Counting110K [52]. The generic VQA data type we developed focuses on enabling the model to produce basic and general QA pairs. We incorporate VQAv2 and GQA, two fundamental VQA datasets for granting LOVA$^3$ the capability to learn how to ask questions like a human. Additionally, to increase the question diversity, we introduce two supplementary VQA tasks: counting VQA and long-response VQA.Counting110K$^†$, a dataset developed in-house by reformulating the original PointQA [52]

Table 9: 665K instruction data of LLaVA1.5. The content is from LLaVA1.5 for convenient browsing.

| Dataset Name | Size | Instruction Prompts |
|---|---|---|
| LLaVA [43] | 158K | – |
| ShareGPT [74] | 40K | – |
| VQAv2 [26] | 83K | Answer the question using a single word or phrase. |
| GQA [30] | 72K | |
| OKVQA [54] | 9K | |
| OCRVQA [56] | 80K | |
| A-OKVQA [71] | 50K | Answer with the option's letter from the given choices directly. |
| TextCaps [75] | 22K | Provide a one-sentence caption for the provided image. |
| RefCOCO[31, 53] | 30K | *Note: randomly choose between the two formats*
Provide a short description for this region. |
| VG [33] | 86K | Provide the bounding box coordinate of the region this sentence describes. |
| Total | 665K | |

Table 10: Hyperparameters of LOVA$^3$ are the same as the LLaVA1.5.

| Hyperparameter | Finetune |
|---|---|
| batch size | 128 |
| learning rate | $2 \times 10^{-5}$ |
| learning rate schedule | cosine decay |
| learning rate warmup ratio | 0.03 |
| weight decay | 0 |
| epoch | 1 |
| optimizer | AdamW |
| DeepSpeed stage | 3 |

format, contributes to the counting type data generation. Moreover, for most of the above generic VQA with a short response, it is necessary for MLLM to learn to ask questions with long answers. Thus, we leverage the conversation subset of LLaVA-150K [43] and then filter out overly lengthy sentences (e.g., the word number $\geq$ 200.) to yield LLaVA-250K$^\dagger$ with almost 250K samples.

**Multi-choice VQA.** Apart from generic VQA, there is another variant known as multi-choice VQA. This format has become increasingly popular in recent multimodal benchmarks, including ScienceQA [50], SEED [36], MMBench [45] and MMMU [93]. Such a data format can be seen as a better way to evaluate the reasoning ability of MLLMs rather than the direct text response in an open-ended way. As each MC VQA sample comprises one correct answer and three incorrect yet plausible alternatives, it will introduce a higher level of complexity for model learning. Thus, We proceed to make the LOVA$^3$ learn to produce multi-choice VQA data.

**Multi-turn VQA.** It is also a complex multimodal data format. We incorporate this data type into the training of LOVA$^3$, enabling it to master the art of generating varied questions within a dialogue context from a single image. Recognizing that the VQAv2 and GQA datasets offer multiple questions per image, we carefully select 83,000 and 72,000 multi-turn VQA instances from each dataset, respectively.

**REC and REG.** Recent studies such as Shikra [10] have recognized the importance of making MLLM talk about the image regions for asking and answering (e.g., describing the region by giving a bounding box in an image) in grounding tasks. Being able to refer to a region precisely when asking or answering a question demonstrates a strong capability of multimodal reasoning. Possessing this capability would enhance an MLLM's potential as an intelligent assistant in human-AI interactions by accurately identifying and referring regions of interest. Thus, besides considering the aforementioned multimodal tasks, we also consider involving grounding tasks like REC and REG to enhance the model capability related to positions. For REC, it aims to ground the region of an image with the given referring expression. About REG, the target is to give the corresponding expression when giving the exact coordinates. We randomly select 30K samples from RefCOCO [31] and Visual Genome (VG) [33], respectively.

**How about the asking ability of LOVA$^3$?** We provide some examples of prompting LOVA$^3$ to generate VQA pairs as in Fig. 9. One can see that LOVA$^3$ is capable of asking versatile questions based on the content of the unlabeled images. Such results demonstrate the potential of the current MLLM to actively ask questions. We would like this finding to inspire future works that explore human-AI interaction in depth.

# D EvalQABench

**Why Fuyu-8B and Llama 2?** To build the EvalQABench, we used two open-source models, Fuyu-8B and Llama 2, to generate negative answers and feedback, respectively. These models were chosen due to the zero financial cost of producing a training dataset. Furthermore, our empirical investigation found that Fuyu-8B and Llama 2 are capable of generating the data by following the instruction prompts described in Sec. 3.2. Such results prove that GPT-4 is not necessary for our purpose.

**Why does manual filtering and correction work?** The reason is that the models of Fuyu-8B and Llama 2 are two closed-formed models, which will output incorrect samples with similar patterns even set by the hyperparameters of inference mode. Therefore, we observe that use manual checking is feasible and enough to remove most of the failure cases. Moreover, GPT-4 is a closed-form model yet that exhibits error patterns.

**Verification of data.** As mentioned in the main paper, we select 100,000 samples from annotated VQA pairs of the VQAv2 training set and then use Fuyu-8B to generate negative answers for subsequent manual filtering and error correction. We obtain 61,094 filtered samples, which is approximately 61% accuracy in generating negative answers. After that, we prompt Llama 2 to produce feedback. In this process, we also manually filter the samples with incorrect formats and finally obtain 41,592 samples. It is almost 68% accuracy in creating feedback.

**Details of each procedure.** In detail, we present the amounts of each procedure in creating the EvalQABench training set in Tab. 11. Initially, we select 100,000 samples and then use Fuyu-8B to obtain 99,998 valid outcomes, and then we use manual filtering to remove 38,904 samples. We conduct error correction to 14,814 samples and then pass 61,094 samples to Llama 2 for feedback generation. After that, we adopt manual filtering to remove 19,502 samples with incorrect formats. The filtering of feedback includes overlength output or none of output.

Table 11: Data amount details of creating EvalQABench training set.

| Procedures | Amount |
|---|---|
| Raw Data | 100,000 |
| Negative answer generation | 99,998 |
| Manual filtering | 61,094 (-38,904) |
| Error Correction | 61,094 (14,814) |
| Feedback generation | 61,094 |
| Manual filtering | 41,592 (-19,502) |

**Data distribution across categories** We provide the data distribution of the nine question types across the "Object", "Yes/No", "Counting", "Color", "Number", "Attribute", "Relation", "Action", and "Others" of the EvalQABench training set in Tab. 12. One can observe that there are 26.7% question belongs to Others. It is noted that "Others" includes versatile questions such as "What does the image represent?", "Who is not out of focus?", "What does the back of the bus say?", "What time does the clock report?", and "How old is this man?". These questions, with diverse scopes, bring diversity to our EvalQABench. Due to the inherent question bias in the original VQAv2 [26] training set, the number of questions categorized as "Number" is limited. Nevertheless, it is important to note that such biases are also present in real-world scenarios. We also provide the statics of the other seven question types in the table.

**Data distribution of negative answers.**

To analyze the data distribution of produced negative answers, we build a word cloud in Fig. 5. One can observe that "Yes" and "No" are the two majority negative answers due to the higher proportion of "Yes/No" questions. While colors and numbers are also the two high-frequency word types that appeared in the negative answers.

**Data distribution of feedback.**

For analyzing the distribution of feedback, we dive into three aspects: sentence length of feedback, noun counts, and verb counts as in Fig. 6, 7, 8.

Table 12: Statistic of question types of EvalQABench training set.

| Statistic | Number | Proportion |
|---|---|---|
| Total Questions | 32,000 | – |
| - Object | 2,418 | 7.55% |
| - Yes/No | 6,804 | 21.26% |
| - Counting | 4,880 | 15.25% |
| - Color | 3,756 | 11.73% |
| - Attribute | 343 | 5.67% |
| - Number | 1,814 | 1% |
| - Relation | 2,380 | 7.44% |
| - Action | 1,274 | 3.98% |
| - Other | 8,331 | 26.03 % |

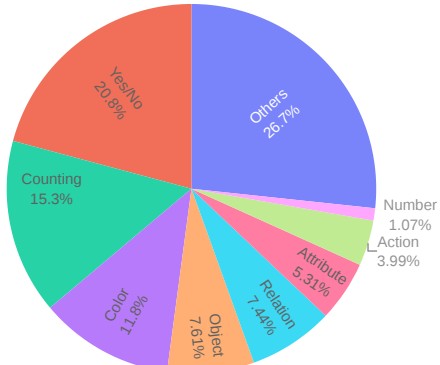

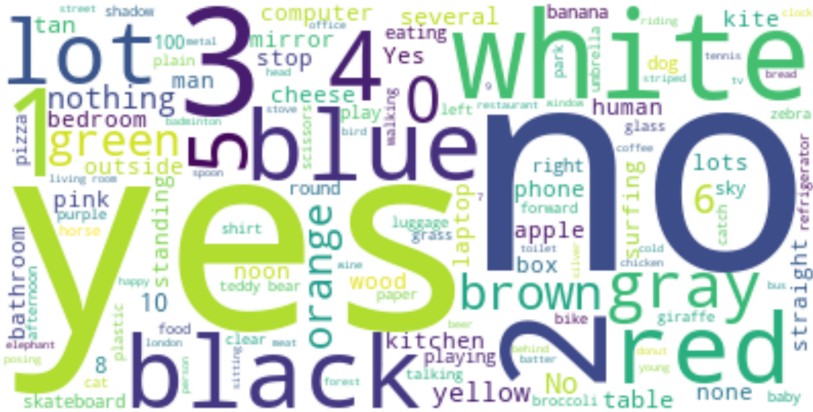

Figure 5: The word cloud of total negative answers.

# E   Failure cases of EvalQABench

In this section, we show some failure cases of our LOVA[3]. As shown in Tab. 13, there are three failure cases: the first case is from the TextVQA [76], and the last two cases are from MM-Vet [92]. In the first case, when the watch is rotated -90° and an incorrect OCR reference is provided, our LOVA[3] fails to give the correct time. The second example highlights the current limitations of LOVA[3] in mathematical calculations and multi-step reasoning. In the third example, LOVA[3] fails to correctly interpret the window of the living room, resulting in an incorrect answer. It is important to note that due to the limited text-centric instruction tuning datasets and mathematic relevant task-specific data in the current experimental settings, LOVA[3] falls short in handling text-centric VQA, and mathematic problem-solving. We believe these failure cases are primarily caused by the shortage of relevant instruction tuning datasets. This leaves room for future exploration, while our work focuses mainly on highlighting the importance and effectiveness of two additional high-level abilities for enhancing multimodal understanding in this paper. We believe that exploring the creation or collection of more mathematical or text-centric data for training will be essential for future work.

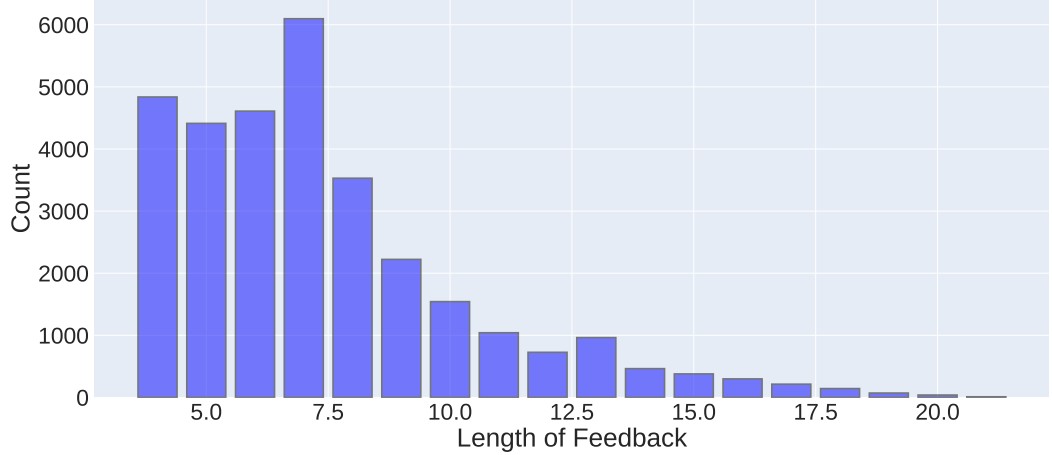

Figure 6: The distribution of the length of feedback.

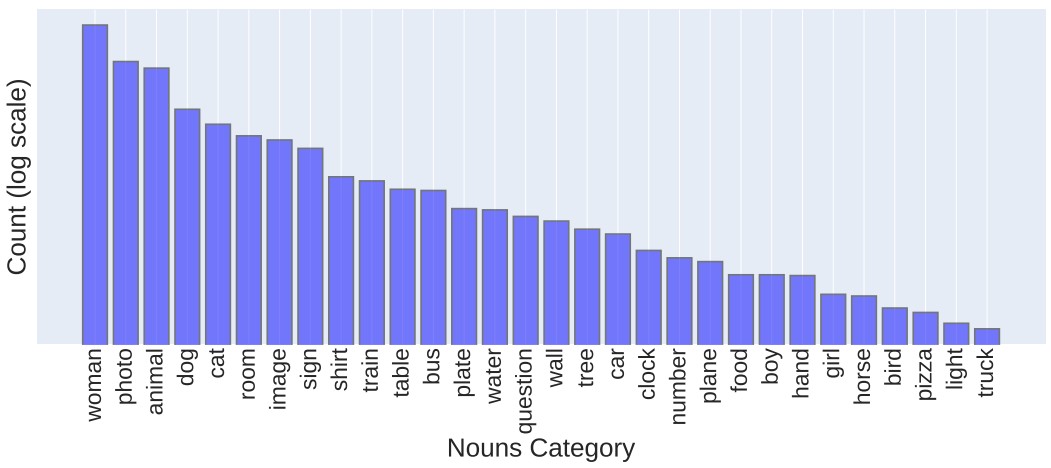

Figure 7: The top-30 nouns of feedback.

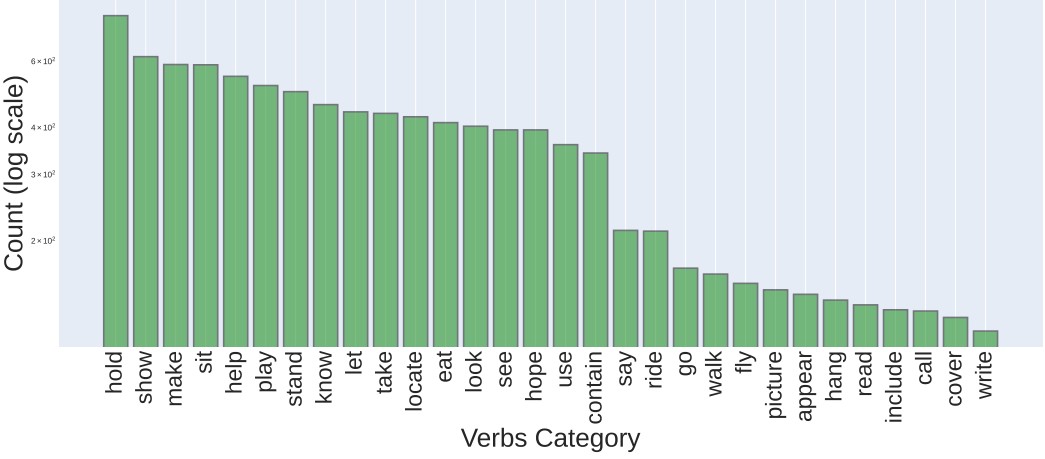

Figure 8: The top-30 verbs of feedback.

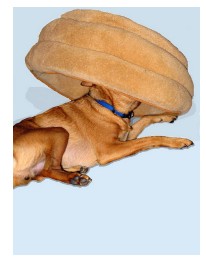

**Question**: What is the dog doing in the image?
**Answer**: The dog is laying down and resting its head on a cushion or a brown pillow.

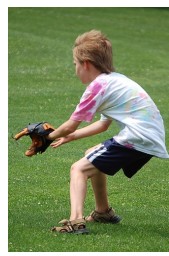

**Question**: What is the boy doing in the image?
**Answer**: The boy in the image is playing baseball, wearing a catcher's mitt, and is in a ready position to catch a ball.

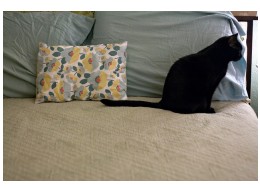

**Question**: What is the cat doing in the image?
**Answer**: The cat is sitting on a bed next to a pillow.

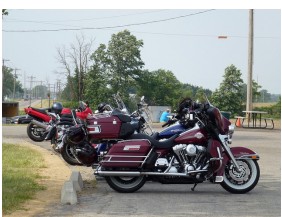

**Question**: How many motorcycles are there?
**Answer**: 4

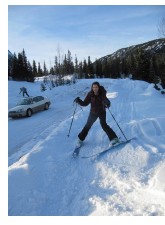

**Question**: Is the woman in the image skiing or snowboarding?
**Answer**: The woman in the image is skiing, not snowboarding.

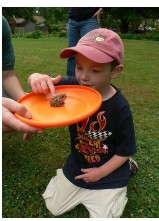

**Question**: What is the boy doing with the frog?
**Answer**: He is either showing the frog to someone or is about to let the frog jump off the plate.

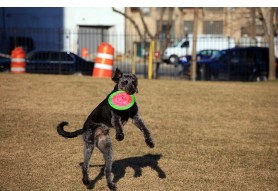

**Question**: What is the dog doing in the image?
**Answer**: The dog in the image is jumping in the air to catch a frisbee in its mouth.

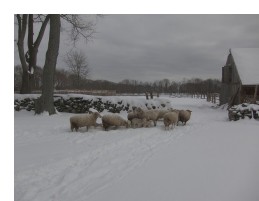

**Question**: What is the animal that is standing in the snow called?
**Answer**: sheep

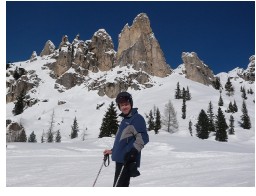

**Question**: What is the man wearing in the image?
**Answer**: The man in the image is wearing a blue jacket and a helmet.

Figure 9: We ask the LOVA[3] to generate the diverse question-answer pairs.

Table 13: Some failure cases of LOVA[3].

Image:

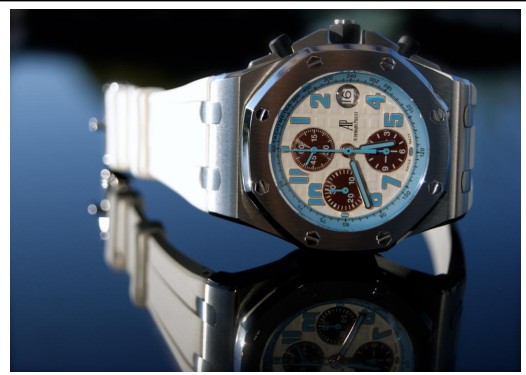

Question: What is the time? \Reference OCR token: N, u, g0 \Answer the question using a single word or phrase.

Answer: 10:10

Image:

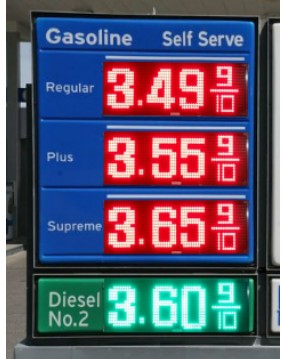

Question: How many gallons of supreme gasoline can I get with $50?

Answer: With $50, you can get approximately 110 gallons of supreme gasoline, as indicated by the gas station sign in the image.

Image:

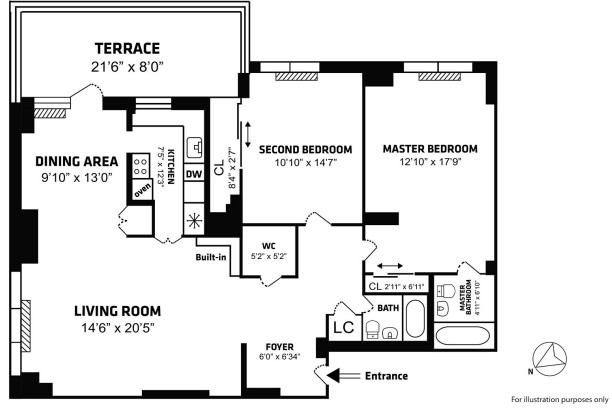

Question: In which direction does the living room window face?

Answer: The living room window faces the dining area.

