# OpenReview forum: "LOVA3: Learning to Visual Question Answering, Asking and Assessment"
_NeurIPS.cc/2024/Conference — NeurIPS 2024 poster_

### Official Review · Reviewer_5aGT · 2024-07-13

**Soundness:** 3
**Presentation:** 3
**Contribution:** 3
**Rating:** 7
**Confidence:** 4

**Summary:**

This paper augments presents a data augmentation / multi-task learning technique to improve model quality for Visual Question Answering (VQA). The key idea of the paper, motivated by analogy to humans, is that asking questions and assessing answers are also key skills, apart from just answering questions. The paper seeks to train a model to "Answer, Assess, Ask" jointly, relying existing datasets for a variety of answering and asking tasks, and deriving a new dataset called EvalQABench for the assessment task. The introduction of the EvalQABench dataset, initially created by LLMs and later filtered by experts, is another potentially valuable and lasting contribution. Multiple tasks on augmented data are implemented on the LLava backbone which is an existing SOTA model. The paper compares their technique (called LOVA) to several SOTA models on a variety of datasets showing robust gains in a multitude of settings, providing confidence in the technique's validity.

**Strengths:**

The paper is well motivated: assessing and asking (evaluation and question generation) are closely associated tasks to question answering, that can be performed on datasets that are easily derived from question answering datasets. The argument that training on these closely related tasks improves generalization on question answering is intuitive though reliant on analogy with humans, which has its own traps.

The paper evaluates their technique against a variety of SOTA models, and across a multitude of tasks, proving that the gains are robust. The paper also provides ablation studies for various components, showing their utility. In general the experiment section is detailed, extensive and is a highlight of the paper.

The paper has 100 citations, and extensive references to related work, making it easier to assess the novelty of the work.

**Weaknesses:**

As the authors point out, due to cost considerations, the authors only evaluate the technique on smaller (relative to MLLMs) models. This is important as model size is a confounder when it comes to assessing the usefulness of data augmentation or multi-tasks. A technique useful for a 7B model is not necessarily useful for a 70B model. However, given the cost of inference of larger models, improving smaller models to be competitive with larger models has its own benefit.

There is prior work that already includes question answering and question generation, for example the InstructBLIP paper. Viewed in that sense, this paper makes an incremental contribution adding the assessing task to the answering, asking combination that was already shown to be useful earlier. However, the EvalQABench dataset is potentially very useful for the whole subfield of visual question answering. One minor but interesting finding in the paper is that in a balanced dataset split rightdown with model with 50% Yes and 50% No answers, not all models predict Yes/No close to 50% of the time.

**Questions:**

In section 1, there's a claim that the EvalQABench datasets is generated via a "new automatic pipeline". However, in section 3.2 the authors say "... acknowledging that the Fuyu-8B model is not flawless and recognizing that no multimodal model, including GPT-4V, is perfect, we have implemented both manual filtering and error correction...". Do the earlier claims about the pipeline being automatic overstate the case? Are they necessary?

Does the feedback add value beyond just rephrasing the answer is a longer sentence. A lot of the feedback seems trivial and already captured in the q,a pair. For e.g. "What does the woman have on her back". "backpack" vs "No, the woman has a backpack on her back".  As another e.g. "What are the people doing?", "motorcycling", vs "No, the people in the picture are motorcycling".

**Limitations:**

Smaller models sizes are an understandable limitation as mentioned in the paper, and referenced earlier in the review.

The introduction of EvalQA dataset and the additional "asses" task is the key incremental contribution of the paper. By looking at rows 4 and 7 in Table 6 that shows ablation studies, one could discern the incremental benefit of EvalQA. The deltas in the scores are somewhat underwhelming (unclear if they are significant).

---

> ### Author Rebuttal · Authors · 2024-08-05
>
> Thank you for your thorough review! We sincerely appreciate your acknowledgment of LOVA3’s motivation, clarity, novelty, effectiveness, and consistent performance gains.
>
> **W1: The model size.**
>
> We sincerely thank your valuable comments. Due to limited GPU resources, it is hard for us to train with the larger model over 7B. Moreover, we found that the results of our model are compatible with LLaVA1.5 (13B) on VQAv2 (80.3 > 80.0), GQA (63.3 = 63.3), MME (1552.7 > 1531.3). This result demonstrates the effectiveness of our training paradigm.
>
> **W2: Training Data used in InstructBLIP.**
>
> We agree that there are question generation datasets in InstructBLIP, but our GenQA has the following differences:
>
> (1) We ask the model to generate questions and answers jointly, not only the question.
>
> (2) We use not only generic data types but also multi-choice VQA, multi-turn VQA, and two grounding data types REC and REG for GenQA task. This increases the difficulty of the GenQA task and enhances the model's problem-solving abilities.
>
> **Q1: Claim of  "New automatic pipeline."**
>
> Thank you for your suggestion. The term "New automatic pipeline" in Section 1 refers to the automatic data generation process and does not include the data filtering process, which could lead to confusion. We will revise this in the next version.
>
> **Q2: "Does the feedback add value beyond just rephrasing the answer in a longer sentence? A lot of the feedback seems trivial …"**
>
> Yes, the feedback is generated by rephrasing the question and answer.
>
> Firstly, we want the model not only to give the classification of "Yes/No", but we would also like there exists a simple explanation. Thus, we use the LLaMA-2 to rephrase the question and the ground-truth answer to generate feedback. During training, the ground-truth answer and negative answer do not appear simultaneously. It is not trivial to add such feedback.
>
> Secondly, rephrasing the ground-truth answer and question is an efficient way to create data. Generating feedback for 32,000 images manually is impractical, so we utilize LLM assistance. Our preliminary study shows that this strategy ensures high accuracy in feedback generation and is effective for training the model to assess the VQA triplet.
>
> **Limitation: About the results of EvalQA in Table 6**
>
> The reason for the incremental results in Table 6 of EvalQA data may be the following:
>
> (1) Data Size: Compared to the other two tasks, VQA (665K) and GenQA (842K), EvalQA includes only 64K data points.
>
> (2) Data ratio: It is crucial for training MLLMs, as highlighted by InstructBLIP and LLaVA1.5. Therefore, the smaller data size of the EvalQA task affects the results in rows 4 and 7 of Table 6. However, referring to row 3 in Table 6, one can observe improvements even with only 64K additional data included.

---

### Official Review · Reviewer_4XZr · 2024-07-13

**Soundness:** 3
**Presentation:** 3
**Contribution:** 2
**Rating:** 4
**Confidence:** 4

**Summary:**

This paper enhances the MLLM's visual understanding capability by training it to ask questions about an image and evaluate the correctness of given question-answer pairs about an image. To achieve this goal, new data is extracted from existing datasets and a new model is fine-tuned on the new data. The experiment shows that the newly added data can improve the MLLM's capability of understanding of images with higher scores on VQA tasks.

**Strengths:**

- The paper is generally well-written and easy to understand.
- The argument that training a MLLM to ask questions and evaluate answers can improve its visual understanding is reasonable and, verified by the well-conducted experiments in the paper.
- The experiment setups are carefully designed to avoid unfair comparisons.

**Weaknesses:**

- The three key capabilities of MLLMs covered by the paper--asking, answering, and evaluation--should be characterized in an interactive environment (like in an embodied environment where the MLLM is treated as the high-level perceiver/planner/controller of robots) instead of in the static environment. Consider, for example, an MLLM doing an embodied task that needs asking about some key questions, this is where the asking capabilities really make sense. However, the paper only trains and evaluates the MLLM in simple VQA problems as in previous literature. In the paper's current state, the value of the paper is limited and, from my perspective, does not meet the bar of acceptance if VQA tasks are considered only. The scope of the paper needs to be increased to a significant extent that touches the essence of MLLMs with higher-level capabilities that incorporate iterative/interactive thinking and planning.

- The added synthesized data only gives the model a limited improvement in performance, while adding a large amount of computation overhead. In fact, if we use models like GPT-4(V) to synthesize random VQA data,  the performance will increase as well [1], so I do not see the clear benefit of specifically doing the asking and evaluation data augmentation. This problem is relevant to the first problem: the capability added to MLLM should not be evaluated in VQA tasks.

[1] Cambrian-1: A Fully Open, Vision-Centric Exploration of Multimodal LLMs.

**Questions:**

(Table 6) Why does adding GenQA-Grounding data improves ScienceQA performance?

**Limitations:**

The paper mentions some limitations of the proposed pipeline. However, as mentioned above, the biggest limitation is the limited scope of the considered setting which only involves VQA (including grounding) problems without considering the embodiment of MLLMs.

---

> ### Author Rebuttal · Authors · 2024-08-06
>
> Thank you for your thoughtful and valuable review. We thank your suggestions about extending the research scope to other domains.
>
> **W1-1: Why do we apply three key capabilities in the current static environment rather than in an interactive environment?**
>
> Firstly, it should be noted that these three abilities are not exclusive to interactive environments such as embodied AI. Other research domains, such as GUI Assistant [12], AI Medical Diagnosis [13], and LLM-Agent [14, 15], also involve asking and assessments in their systems. Both static and interactive environments require the incorporation of these three capabilities. Therefore, incorporating these abilities into MLLM model training can be recognized as one strength of our work that other MLLMs neglect.
>
> Secondly, the scope of this research is about training MLLM. We are motivated by the current model training of MLLMs that primarily focuses on teaching models to answer the questions while neglecting the asking and assessing abilities. Our investigation of current MLLMs, including LLaVA, Qwen-VL [16], and CogVLM [17], revealed that while these models excel at answering questions, they perform inadequately in asking and assessing skills. Therefore, we consider that it is necessary to explore adding asking and assessing abilities in training MLLM. Our experiments in Tables 3, 4, and 5 reconfirm that these three abilities actually enhance models’ problem-solving capabilities.
>
>
> **W1-2: Is it appropriate to say that the asking capability of an MLLM makes more sense in embodied tasks?**
>
> (1) We agree that asking key questions is important for embodied tasks. However, other tasks, such as AI diagnosis and AI assistant systems, also require this ability. It is not exclusive to embodied tasks. In other words, this ability is applicable not only to embodied tasks but also to other domains and is worth exploring in MLLM training.
>
> (2) We assume that if an MLLM is able to yield high-quality questions and corresponding answers, it indicates a stronger problem-solving ability and a deep visual understanding. This is similar to human cognition: after thoughtfully understanding concepts, we can well ask questions. Refer to the ablation study in Table 6, by adding GenQA data, there are consistent improvements in VQA datasets. Therefore, we consider adding asking tasks to be essential in MLLM training.
>
> (3) Highlighting the importance of asking ability is also the affirmation of our contribution about incorpararting GenQA in the model training, which other MLLMs have neglected.
>
> **W1-3: Is it acceptable for testing on VQA and multimodal benchmarks?**
>
> Firstly, we follow the previous representative MLLMs, such as LLaVA1.5, InstructBLIP, Qwen-VL, and CogVLM, to test our model on widely used VQA datasets and multimodal benchmarks so that we can directly compare our results with theirs.
>
> Secondly, VQAv2, GQA, VizWiz, ScienceQA, and the object hallucination VQA dataset POPE are widely used VQA datasets for testing MLLMs. Additionally, we select five popular multimodal benchmarks—MME [18], SEED-Bench, MMBench [19], LLaVA-Bench, and MM-Vet [20]—which encompass a wide range of diverse and challenging multimodal questions across various domains (e.g., Math, Instance Attribute, Spatial Relation, Text Understanding, Object Recognition, Scene Understanding) especially for testing MLLMs.
>
> Thirdly, we appreciate your suggestion to extend the scope of our research to embodied AI and consider it a potential area for future exploration.
>
> **W2-1: The added synthesized data only gives the model a limited improvement in performance while adding a large amount of computation overhead.**
>
> (1) The improvement is not limited, as we chose LLaVA1.5 (7B) as the baseline and trained the model following the original training recipe without tuning any hyperparameters. Additionally, we did not involve any new images in training. Therefore, the clear improvements shown in Tables 3, 4, and 5 are meaningful. Some results of our 7B model are even compatible with the larger model LLaVA1.5 (13B):
>
>
> | | Size | VQAv2 | GQA | VizWiz | MME |
> | --- | --- | --- | --- | --- | --- |
> | LLaVA1.5 | 7B | 78.5 | 62.0 | 50.0 | 1510.7 |
> | LLaVA1.5 | 13B | 80.0 | 63.3 | 53.6 | 1531.3|
> | LOVA$^3$ | 7B | 80.3 | 63.3 | 53.6 | 1552.7 |
>
> (2) The training costs are not high. We trained our model for only 24.5 hours on an 8 A100 (40G) GPU setup.
>
> (3) As shown in Table 6, the first six rows, which present results with different data sizes, consistently outperform the baseline model LLaVA1.5. This indicates that our training paradigm performs well even with smaller data sizes, including as low as 64K data points in EvalQA.
>
> **W2-2: If we use models like GPT-4(V) to synthesize random VQA data, the performance will increase as well.**
>
> Synthesizing data to improve performance is non-trivial and is crucial for training MLLMs. The key challenge is how to generate essential data at a low cost. We propose a solution that does not require GPT-4(V) or new images. Instead, it leverages existing annotations to achieve a significant performance improvement. For the GenQA task, the training data is derived from the original training datasets used in LLaVA1.5, such as VQAv2 and GQA. For the EvalQA task, we use VQAv2 as the source for data generation. As shown in the ablation study in Table 6, there are robust gains by adding GenQA and EvalQA data. These results indicate the effectiveness of our created data.
>
> **Q1: Why does adding GenQA-Grounding data improve ScienceQA performance?**
>
> (1) Training with GenQA-Grounding data enhances the ability to understand object positions deeply, which is beneficial for fully leveraging visual information for reasoning.
>
> (2) Many images in ScienceQA [21] (https://scienceqa.github.io/)  are from natural scenes. Therefore, enhanced visual understanding would improve the accuracy of reasoning is reasonable.

---

> > ### Comment · Reviewer_4XZr · 2024-08-14
> >
> > Thanks for the response.
> >
> > Regarding your first, second, and third arguments, other agents, like GUI Assistant/digital-device agent and LLM-Agent still need an interactive environment in the loop, so you still need to change your evaluation settings. My opinion is not restricted to robotics tasks. It is not appropriate to say that "previous works use VQA tasks for evaluation, so we use them as well"; a good paper should choose/create the most appropriate settings that can validate the main claims.
> >
> > I keep the score.

---

> ### Author Response · Authors · 2024-08-12
> **Official Comment by Authors**
>
> Dear reviewer, we greatly appreciate your time in reviewing our response. We hope that your concerns are addressed with our rebuttal. Please let us know if there are any further questions that need clarification.

---

> ### Author Response · Authors · 2024-08-13
> **Author Response to Reviewer 4XZr**
>
> Dear Reviewer 4XZr,
>
> We deeply appreciate the time you have devoted to reviewing our manuscript. In our rebuttal, we have carefully considered and addressed your concerns. As we have not yet received your feedback during the discussion period, we would like to summarize our rebuttal below to help clarify the key points.
>
> Regarding your main concerns about evaluating the answering, asking, and assessment capabilities on generic multimodal tasks rather than embodied tasks, we would like to express our viewpoint:
>
> - **Multimodal instruction tuning is a crucial and foundational area that can significantly benefit downstream tasks, including embodied tasks and GUI assistance.**
>
> - **We would like to emphasize that evaluating MLLMs on VQA tasks is a widely accepted practice, as demonstrated by other models such as LLaVA1.5, Qwen-VL, and CogVLM. This evaluation is essential to assessing the overall capabilities of MLLMs.**
>
> - **Applying questioning and assessment abilities is equally important in both general MLLM research and embodied AI research. Moreover, our experiments reaffirm that adding the two additional abilities is able to bring consistent and robust gains. Therefore, conducting research with MLLMs tested on VQA tasks is a non-trivial and essential endeavor.**
>
> We trust these responses effectively address the concerns you raised. As the discussion period deadline approaches, we eagerly await any further feedback you may have.
>
> Once again, thank you for your dedication to the review process.
>
> Best regards,
>
>
> Authors of Paper 9008

---

> ### Author Response · Authors · 2024-08-14
> **Author Response to Reviewer 4XZr**
>
> Dear Reviewer 4ZzR,
>
> Firstly, there is no need to change our current settings. **There is no clear evidence to suggest that asking and assessing are exclusively applicable to the interactive environment.** And then, we demonstrate that incorporating the two additional tasks is **benefit** for MLLM training. The tasks, such as embodied AI and GUI Assistant, are **downstream tasks**, whereas our focus is on **multimodal foundation model training**. Adapting to these tasks would require **entirely different methods and experiments, essentially resulting in a separate paper**. It is important to clarify that it is not inappropriate to focus on downstream tasks before evaluating the approach on a foundation model.
>
> Secondly, **we referred to previous works to underscore the significance of multimodal instruction tuning and to provide evidence supporting the validity of evaluating on VQA datasets and benchmarks.** These examples serve as the foundation of our argument. We respectfully disagree with your viewpoint, particularly since the **other two reviewers found our experimental setup to be reasonable, fair, and insightful.**
>
> Thirdly, as shown in Table 7, we have already evaluated the assessing ability in comparison with other SOTA models. The results clearly demonstrate that our model significantly enhances this ability without introducing prediction bias. Our evaluation setting was conducted reasonably.
>
> Best regards,
>
> Authors of Paper 9008

---

> > ### Comment · Reviewer_4XZr · 2024-08-14
> >
> > Thanks for the response. I understand that your focus is on multimodal foundation model training, but if your method is general and most likely to be useful for some specific task (here interactive agents), the evaluation should better reflect this point.  I am not saying that the proposed method is useless, but that the paper can be largely improved and put into a more appropriate context if evaluation settings change from VQA to interactive tasks. The authors are encouraged to do so in order to make the paper much stronger to have a larger impact on the community.
> >
> > On the other hand, the authors said that "Adapting to these tasks would require entirely different **methods** and experiments", but if your method is to improve the general capability of foundation models, why do you need entirely different **methods** when considering that the improved capability is closely related to the tasks?
> >
> > Considering that the authors have clarified some of the points, I raise the score to 4.

---

### Official Review · Reviewer_UkuU · 2024-07-14

**Soundness:** 2
**Presentation:** 3
**Contribution:** 2
**Rating:** 3
**Confidence:** 4

**Summary:**

The paper introduces LOVA3, a framework designed to enhance Multimodal Large Language Models (MLLMs) by incorporating not only visual question answering (VQA) but also the capabilities of generating questions (GenQA) and evaluating question-answer pairs (EvalQA). The primary objective is to improve the comprehensive multimodal understanding of AI models.

LOVA3 includes the development of EvalQABench, a benchmark with 64,000 training samples to evaluate VQA data quality. The framework uses the LLaVA-1.5 model as a base, incorporating datasets like VQAv2 and GQA to train these new tasks. Experimental results on ten multimodal benchmarks demonstrate that LOVA3 significantly improves the models' performance, highlighting the benefits of incorporating comprehensive questioning and evaluation abilities into MLLMs. The paper emphasizes the approach and robust results, despite noting the increased computational cost and the need for further testing on larger models and domain-specific tasks.

**Strengths:**

1. LOVA3 introduces a strategy that extends beyond traditional VQA tasks by incorporating question generation and evaluation.
2. The creation of EvalQABench provides a rigorous way to test and improve MLLMs.
3. The multiple perspectives of experimental results provide insights of the proposed framework across multiple benchmarks.

**Weaknesses:**

1. Incorporating additional tasks like GenQA and EvalQA, but the two tasks are also the existing steps of the visual language instruction generation for visual question answering (e.g. SEED-Bench) or visual instruction tuning (e.g., LLaVa-Bench). They also used LLMs or MLLMs for the dataset generation and validation. To explained the special novelty or contribution would be better.
2. The work doesn't provide detailed explanations on how to validate the generated data quality from humans instead of using imperfect models (LLMs or VLMs).  It uses Fuyu-8B for data generation but employs a stronger MLLM (LLaVA 1.5) as the base model for instruction tuning. Since LLaVA 1.5 is stronger than Fuyu-8B, the generated negative samples would be less challenging and easier to recognize by stronger models.
3. The paper lacks a more in-depth analysis of potential data biases and strategies to mitigate them.
4. The proposed benchmark is relevant to visual question answering and data generation for visual question answering. It would be necessary to survey and discuss the recent existing datasets (e.g., VQA-GEN, CrossVQA, OVQA, STAR) and generated benchmarks (e.g., LLaVA-Bench, SEED-Bench, SOK-Bench, CinePile) as fully considered.
5. The paper does not provide the generated dataset for review, which is important for the validation of the work.

**Questions:**

1. How about the prompt stability of the QA generation and the differences when using the different variants of prompts?
2. Why does the work apply Fuyu-8B instead of LLaVA 1.5 for the data generation and is there any comparison between the different new VLMs?

---

> ### Author Rebuttal · Authors · 2024-08-05
>
> Thank you for thoroughly reviewing our work! We have carefully considered all your concerns and addressed them in the following responses.
>
> **W1: Comparison with SEED-Bench and LLaVA-Bench in using LLMs or MLLMs.**
>
> (1) GenQA and EvalQA are two new training tasks, whereas SEED-Bench and LLaVA-Bench are test benchmarks. For GenQA, no LLMs or MLLMs are used for data creation. We propose an efficient strategy for creating the data from existing datasets (see Appendix C). For EvalQA, we use Fuyu-8B [3] and LLaMA-2 [4] for data creation. This cannot be a weakness of our work, as creating over 32,000 samples for training the assessment ability necessitates using LLMs/MLLMs rather than human labeling. Additionally, we demonstrate the effectiveness of using non-commercial LLMs/MLLMs in data generation rather than commercial models like GPT-4(V).
>
> (2) Novelty: To the best of our knowledge, LOVA$^3$ is the first work to imbue the asking and assessment abilities in training a robust and intelligent MLLM. Additionally, we are the first to propose GenQA and EvalQA tasks for enhancing the model’s problem-solving ability. Contribution: We contribute a new training pipeline, a new benchmark EvalQABench, and an open-source model.
>
> **W2-1: Is LLaVA1.5 stronger than Fuyu-8B?**
>
> It is not appropriate to claim that LLaVA1.5 [5] is stronger than Fuyu-8B. As Fuyu-8B is trained with large-scale VL data, LLaVA1.5 is only trained with about 1.4M data in the whole two-stage training. The zero-shot results provided by Fuyu-8B (https://www.adept.ai/blog/fuyu-8b) show that even without training with the VQAv2 and GQA datasets (training data of LLaVA1.5), Fuyu-8B achieves exceptional performance on multiple VQA datasets.
>
> **W2-2: Are the generated negative samples easier to recognize or of lower quality?**
>
> Firstly, the generated samples are not simple for model training. As shown in Table 7, these state-of-the-art (SOTA) MLLMs still suffer from inferior accuracy on the generated data, indicating the generated data remains challenging.
>
> Secondly, LLaVA1.5 is prone to predicting “Yes” (nearly 66%), as shown in Table 7, indicating that the EvalQABench samples are not easy for LLaVA1.5 to identify.
>
> Thirdly, the EvalQA task differs from the VQA task. If we use Fuyu-8B or GPT-4 to build VQA synthetic datasets and then use the synthetic data to train the model, it may lead to model degradation. However, we only use Fuyu-8B to produce negative answers for constructing negative samples for model assessment. Thus, EvalQA data would not restrict the VQA performance.
>
> **W3: In-depth analysis of potential data biases.**
>
> (1) For GenQA, we build on existing annotated datasets from the original LLaVA1.5 training data. Since we incorporate not only generic VQA data but also Multi-Choice VQA, Multi-turn VQA, and grounding like REC and REG, our GenQA dataset is diverse. As shown in Table 1, we strictly follow the data ratio of LLaVA1.5 for each data type. Thus, our experimental settings have no obvious data biases in terms of data types and amounts.
>
> (2) For EvalQA, we use the VQAv2 dataset, a subset of the original training corpus of LLaVA1.5. Therefore, no new datasets are used. Moreover, VQAv2 is already a balanced version of VQAv1. We create the EvalQABench train set by randomly selecting 100,000 samples from the VQAv2 train set and finally yielding 32,000 negative samples (see Appendix D). As shown in Table 2, we have 9 question types of EvalQA tasks, for each type, with manually restricted ratios for data balance.
>
> **W4: Comparison with other datasets or benchmarks.**
>
> Thanks for your suggestion regarding the comparison with other datasets. Here are the unique aspects of GenQA and EvalQA:
>
> (1) Unlike the traditional Visual Question Generation (VQG) task, which focuses primarily on the generic VQA domain, GenQA is designed to generate diverse VQA data. Additionally, GenQA generates both questions and answers simultaneously, whereas traditional VQG focuses solely on question generation.
>
> (2) EvalQABench is designed to assess VQA data rather than answer VQA questions. In contrast, other VQA benchmarks (e.g., VQA-GEN [6], CrossVQA [7], OVQA [8], STAR [9]) and multimodal benchmarks (e.g., LLaVA-Bench, SEED-Bench, SOK-Bench [10], CinePile [11]) primarily evaluate a model's ability to answer questions.
>
> Compared to the datasets you mentioned:
>
> (1) The data generation process of CrossVQA contains the joint question and answer generation. However, our GenQA is considered an additional task for enhancing the model's comprehension ability, not a test dataset for assessing distribution shifts.
>
> (2) OVQA and STAR are video VQA datasets that focus on question answering.
>
> (3) SOK-Bench (May 15, 2024) and CinePile (May 14, 2024) are contemporary works that use ChatGPT for data generation. In contrast, our EvalQABench uses only non-commercial models.
>
> Thank you for recommending these datasets. We will add the comparison in the next version of Section 2.
>
> **W5: The paper does not provide the generated dataset for review.**
>
> We create an anonymous link at https://anonymous.4open.science/r/LOVA3-9008/README.md due to the double-blind policy containing all our created datasets of EvalQABench. Please refer it for further details.
>
> **Q1: The prompt stability of the QA generation**
>
> We train the GenQA task by randomly choosing one from 58 prompts for each data sample along with a short description for the data type, such as “Please provide a clear and direct question and answer after examining the image. This is a Multi-choice VQA task.” Thus, it is stable for QA generation when one prompt is randomly chosen during inference.
>
> **Q2: Why apply Fuyu-8B for the data generation?**
>
> We chose Fuyu-8B because it is a fast, open-source MLLM with exceptional performance on many complex tasks. Through our preliminary study, we found that Fuyu-8B is stronger than LLaVA1.5 with fewer hallucination issues and better visual input understanding ability.

---

> ### Author Response · Authors · 2024-08-12
> **Official Comment by Authors**
>
> Dear reviewer, we would like to thank you for your insightful feedback. We hope that your questions are addressed with our rebuttal. Please let us know if there are any further questions that need clarification.

---

> ### Author Response · Authors · 2024-08-13
> **Author Response to Reviewer UkuU**
>
> Dear Reviewer UkuU,
>
> Thank you again for your valuable reviews of our submission. As we have not yet received your feedback on our rebuttal in the current discussion period, we would like to summarize our key points below to help address your concerns.
>
> - **Regarding the use of LLMs or MLLMs in creating the EvalBench, we believe this approach is non-trivial and essential for training MLLMs effectively.** We would like to clarify that, unlike SEED-Bench and LLaVA-Bench, we did not use commercial models like ChatGPT for data generation. **It brings insights for other future works for their data generation with low financial costs.** Additionally, **current published works LLaVA [1] (NeurIPS 2023), Ferret [2] (ICLR 2024), SNIFFER [3] (CVPR 2024), Next-GPT [4] (ICML 2024), ShareGPT4V [5] (ECCV 2024) also use GPT-4(V) to build their corresponding data** for training specialized MLLMs. It is the common practice of using stronger LLMs/MLLMs for data generation.
>
> - **While we respectfully disagree with the viewpoint that 'LLaVA1.5 is stronger than Fuyu-8B,'** we acknowledge that LLaVA1.5 achieves exceptional performance on diverse datasets and benchmarks. However, this does not necessarily indicate that it is superior to Fuyu-8B, as LLaVA1.5 was trained on foundational and relevant datasets like VQAv2 and GQA. Therefore, it is reasonable to use Fuyu-8B to generate negative answers while the results in Table 6 demonstrate that our synthetic data is high quality which is challenging for SOTA MLLMs.
>
> - Adhere to the policy of NeurIPS, we create an anonymous link **https://anonymous.4open.science/r/LOVA3-9008/README.md that includes all our training and evaluation codes, created datasets, and pre-trained weights**.
>
> We hope these clarification adequately address your initial questions. As the discussion period deadline approaches, we keenly await your further feedback you may have.
>
> Once again, thank you for your dedication to the review process.
>
> Best regards,
>
> Authors of Paper 9008
>
> ---
>
> References:
>
> [1] Visual Instruction Tuning
>
> [2] Ferret: Refer and Ground Anything Anywhere at Any Granularity
>
> [3] SNIFFER: Multimodal Large Language Model for Explainable Out-of-Context Misinformation Detection
>
> [4] NExT-GPT: Any-to-Any Multimodal LLM
>
> [5] ShareGPT4V: Improving Large Multi-modal Models with Better Captions

---

### Author Response · Authors · 2024-08-05
**Author response to all reviewers**

**We sincerely thank all the reviewers for their valuable time and their thoughtful comments and questions. We are encouraged that the reviewers find that**:

- We introduce LOVA$^3$, which enhances MLLM training by incorporating two additional capabilities: question-asking and assessment (UkuU, 4XZr, 5aGT), drawn from human learning mechanisms. This approach is **insightful** (UkuU), **reasonable** (4XZr), and **well-motivated** (5aGT).
- We have developed a novel benchmark for evaluating the accuracy of VQA triplets, featuring carefully selected question types that **bring valuable and lasting contributions** (5aGT). The creation of EvalQABench provides a **rigorous way to test and improve MLLMs** (UkuU).
- We evaluate LOVA$^3$ against a variety of SOTA models across multiple tasks, demonstrating **robust improvements** (UkuU, 4XZr, 5aGT). The results offer **valuable insights** (UkuU) and are **reasonable** (4XZr). The ablation studies for various components **showcase their effectiveness** (5aGT). The experimental settings are **carefully designed and fair** (4XZr).


To better review the details of the proposed EvalQABench and validate the reproducibility of this work, **we create an anonymous link at **https://anonymous.4open.science/r/LOVA3-9008/README.md** following the double-blind policy**. This link contains all the datasets we created for EvalQABench, as well as the training and evaluation codes and model weights. Please refer to it for further details.


Finally, **we would like to emphasize again that our intention with this paper is to advance the multimodal instruction tuning of current MLLMs by introducing two intuitive and reasonable tasks: GenQA and EvalQA along with traditional VQA.** These additions are designed to enhance problem-solving capabilities. By focusing on these three tasks, **we demonstrate significant improvements across diverse benchmarks and offer insights** that can inspire further exploration and development in the field of multimodal instruction tuning. We trust that **our contributions will be thoroughly evaluated by the reviewers, as this training paradigm is genuinely novel** within the current multimodal instruction tuning domain.

We attempted our best to address the questions as time allowed. We believe the comments have made the paper stronger and thank all the reviewers for their help. Please find individual responses to your questions below.


### References:

[1] SEED-Bench: Benchmarking Multimodal LLMs with Generative Comprehension

[2] Visual Instruction Tuning

[3] Fuyu-8B: A Multimodal Architecture for AI Agents

[4] Llama 2: Open Foundation and Fine-Tuned Chat Models

[5] Improved Baselines with Visual Instruction Tuning

[6] VQA-GEN: A Visual Question Answering Benchmark for Domain Generalization

[7] CrossVQA: Scalably Generating Benchmarks for Systematically Testing VQA Generalization

[8] Open-Vocabulary Video Question Answering: A New Benchmark for Evaluating the Generalizability of Video Question Answering Models

[9] STAR: A Benchmark for Situated Reasoning in Real-World Videos

[10] SOK-Bench: A Situated Video Reasoning Benchmark with Aligned Open-World Knowledge

[11] CinePile: A Long Video Question Answering Dataset and Benchmark

[12] AssistGUI: Task-Oriented Desktop Graphical User Interface Automation

[13] MEDIQ: Question-Asking LLMs for Adaptive and Reliable Clinical Reasoning

[14] Reflexion: Language Agents with Verbal Reinforcement Learning

[15] CRITIC: Large Language Models Can Self-Correct with Tool-Interactive Critiquing

[16] Qwen-VL: A Versatile Vision-Language Model for Understanding, Localization, Text Reading, and Beyond

[17] CogVLM: Visual Expert for Pretrained Language Models

[18] MME: A Comprehensive Evaluation Benchmark for Multimodal Large Language Models

[19] MMBench: Is Your Multi-modal Model an All-around Player?

[20] MM-Vet: Evaluating Large Multimodal Models for Integrated Capabilities

[21] Learn to Explain: Multimodal Reasoning via Thought Chains for Science Question Answering

---

### Comment · Area_Chair_Gkg3 · 2024-08-12
**Please read the author rebuttal, other reviews and respond to the authors NOW!**

Dear Reviewers,

Thanks to those of you who already responded to the authors acknowledging the rebuttal and asking follow-up questions if any.

Those who have not responded yet, please do the following ASAP: thoroughly read the rebuttal, the other reviews and respond to the authors about whether all your questions / concerns have been addressed or not. If not, please elaborate on which questions / concerns are still not addressed so that the authors have fair chance of addressing them before the author-reviewer discussion period ends in ~41 hours from now (August 13th, 11:59pm AoE).

Your AC

---

### Decision · Program_Chairs · 2024-09-25

**Decision:**

Accept (poster)

**Comment:**

The paper received divergent ratings of reject (R1), borderline reject (R2) and accept (R3) post rebuttal and post the discussion between the authors and reviewers and the discussion amongst the reviewers. After carefully reading the paper, the reviews, the author rebuttal and all the discussion, the AC recommends accepting the paper because they find the proposed idea of using two additional training tasks of asking and assessing question-answer pairs on top of the standard VQA training task to be novel and interesting -- the paper shows that the generalization abilities of multimodal LLMs such as LLaVA improves significantly (significant performance improvements shown across 10 benchmarks!) when it is trained with two auxiliary tasks on top of the standard visual question answering task -- 1) generating question-answer pairs for given images, 2) judging the correctness of answer given questions and images. Moreover the training datasets for these additional tasks are obtained from datasets that are already used to train the base LLaVA model, thus validating that the performance improvements brought about the proposed method are not merely due to using additional data. The proposed methods leads to significant performance improvements across 10 benchmarks validating the efficacy of the proposed method.

Thus, the paper presents a new training recipe for training multimodal LLMs which does not require any additional images and requires minimal additional annotations (only to obtain negative answers for the EvalQA task) and leads to significant performance improvements across various benchmarks. Thus the proposed method is easy to adopt and can significantly benefit the community by providing a new training recipe for building stronger multimodal LLMs.

The AC is not convinced by R1's and R2's reasons for rejection. Below are the AC's thoughts for each of R1's and R2's outstanding concerns post-rebuttal and post-discussion:

**R1**

> "The response claims that the contributions include proposing two novel tasks (GenQA and EvalQA), but it lacks a clear explanation of how these tasks differ from existing data generation or evaluation tasks used in QA or vision-language contexts. Without concrete distinctions, the novelty of these contributions is unclear."

Authors state that their novelty lies in proposing these tasks as additional tasks for training while using the existing datasets. The AC is convinced with this response from the authors.

> "Why does the work apply Fuyu-8B instead of LLaVA 1.5 for the data generation"

The AC is not worried about this because despite using Fuyu for data generation, which is supposedly weaker than LLaVA as per R1, the proposed method brings about performance improvement in LLaVA.

> "While Table 1 lists existing relevant datasets and their respective sizes, it does not justify why the sum of the ratios presented should be considered an ideal data distribution."

Authors say they follow the data distribution used for training LLaVA.

> "Additionally, no data distributions are provided in Table 2 (which contains only a few examples) or in Appendix D (which details the data amounts at various processing stages). This omission makes it difficult to evaluate the appropriateness of the proposed data distribution -- what data distribution."

These are examples from EvalQA for which authors state they use VQA v2.


**R2**

> "but if your method is general and most likely to be useful for some specific task (here interactive agents), the evaluation should better reflect this point."

The AC is convinced with the author response on this point. The AC agrees that evaluating on VQA tasks only is a useful and sufficient evaluation given the scope of the paper. The evaluation suggested by R2 in embodied environments appears to be out of scope for the paper.

> "The added synthesized data only gives the model a limited improvement in performance, while adding a large amount of computation overhead."

Although the authors provided the training time for the proposed method, for precise comparison, the AC recommends the authors to also provide the training time for the LLaVA baseline.

> "If we use models like GPT-4(V) to synthesize random VQA data, the performance will increase as well [1], so I do not see the clear benefit of specifically doing the asking and evaluation data augmentation."

The AC is convinced with the author response on this that the benefit of the approach is that it makes use of existing datasets without needed additional annotations. Although the EvalAI task does require additional annotations (for negative answers), as shown in Table 6, the proposed method improves over the baseline even in the absence of this task. So it is clear that the performance improvement is not solely due to additional annotations.

Overall, the AC thinks the proposed method would be of value to the community and hence recommends accepting the paper. The AC recommends the authors to incorporate the additional clarifications and experimental results provided in the rebuttal in the camera-ready version of the paper.

The AC is also convinced with the author response to the Ethics reviews but recommends the authors to provide the precise error patters that were automatically detected and filtered out in the filtering stage of curating the EvalQA training dataset.